# Cortical beta power reflects decision dynamics and uncovers multiple facets of post-error adaptation

Adrian G. Fischer [1,2,3], Roland Nigbur[1], Tilmann A. Klein[1,4], Claudia Danielmeier [5] & Markus Ullsperger [1,3]

Adapting to errors quickly is essential for survival. Reaction slowing after errors is commonly observed but whether this slowing is adaptive or maladaptive is unclear. Here, we analyse a large dataset from a flanker task using two complementary approaches: a multistage drift-diffusion model, and the lateralisation of EEG beta power as a time-resolved index of choice formation. Fitted model parameters and their independently measured neuronal proxies in beta power convergently show a complex interplay of multiple mechanisms initiated after mistakes. Suppression of distracting evidence, response threshold increase, and reduction of evidence accumulation cause slow and accurate post-error responses. This data provides evidence for both adaptive control and maladaptive orienting after errors yielding an adaptive net effect – a decreased likelihood to repeat mistakes. Generally, lateralised beta power provides a non-invasive readout of action selection for the study of speeded cognitive control processes.

[1] Institute of Psychology, Otto-von-Guericke University, D-39106 Magdeburg, Germany. [2] Center for Behavioral Brain Sciences, D-39106 Magdeburg, Germany. [3] Department of Education and Psychology, Freie Universität Berlin, D-14195 Berlin, Germany. [4] Department of Neurology, Max Planck Institute for Human Cognitive and Brain Sciences, D-04103 Leipzig, Germany. [5] School of Psychology, University of Nottingham, Nottingham NG7 2RD, UK. These authors contributed equally: Adrian G. Fischer, Roland Nigbur. Correspondence and requests for materials should be addressed to A.G.F. (email: adrian.fischer@fu-berlin.de)

The ability to quickly adapt behaviour to unforeseen outcomes is essential to survival in many ecological environments. Error commission is known to elicit strong neural responses[1] and trigger immediate changes in cognitive control[2]. These adjustments may consist of increases in reaction times (RT) (post-error slowing, PES), a post-error increase in accuracy (PIA), and a decrease in the RT increase caused by distraction (post-error reduction of interference, PERI[3]). These phenomena are assumed to be general mechanisms of performance adjustments because they occur in diverse tasks[4–6]. However, while many studies confirmed performance increases following error commission[7–9], others found performance decreases[10–12].

Mechanistically, adaptive error processing accounts suggest that information processing is delayed following errors[13], motor inhibition is increased[14], or that the response threshold (the evidence required to trigger a response) is augmented, resulting in a speed-accuracy trade-off towards accuracy[15]. Furthermore, adaptive accounts suggest focusing of selective attention specifically to task-relevant information[16]. In contrast, maladaptive error processing has been explained by an orienting response triggered by mistakes: infrequent errors capture attention and deteriorate consecutive performance[17]. The orienting reflex[18] comprises central and autonomic nervous systems' activity and is evoked by unexpected, salient events. It suppresses ongoing motor activity and disengages selective attention from the current focus. This debate is ongoing for several reasons. Neurophysiological evidence in support of either theory is sparse[19–21], because post-error adjustments occur at very fast time-scales and their study requires specifically tuned experimental paradigms. Additionally, no neural measure has been established that allows to track action planning and execution at a sufficient temporal resolution and at the same time reflects the influence of factors driving behaviour, including the direction and degree of distraction on a trial-wise level[2]. Finally, post-error adaptations have rarely been mapped onto formal decision models[22].

Sequential sampling models such as the drift-diffusion model (DDM) assume that decisions are triggered when a decision-variable crosses a response threshold[23]. The decision-variable reflects accumulated evidence and thus threshold increases are associated with slow but accurate responses. Distractors should divert the decision signal from the correct response, and the extent of this effect could serve to assess post-error adjustments. Maladaptive and adaptive accounts make distinct predictions about error-induced changes in response threshold and speed of evidence accumulation. Motor inhibition consecutive to an orienting reflex is implemented by reduced corticospinal excitability[24]. In the DDM framework, this corresponds to boundary increases: the lower cortical excitability, the more evidence is required to trigger a response. If the orienting response triggers rapid disengagement of selective attention from task-relevant stimuli[10,25], this should lead to slower evidence accumulation which could be reflected in lower drift rates in the DDM following errors[20]. Finally, the orienting reflex could also prolong stimulus processing reflected in the DDM's non-decision time. The adaptive error adaptations PES and PIA could mechanistically be implemented in two different ways: either increased motor threshold and/or increased focus of attention. The former would result in higher decision thresholds, the latter in lower drift rates for distractors.

It has recently been found that the motor cortex continually samples information for and against a response and may be involved in the decision-making process itself[26,27]. Signals over motoric brain regions are continuously reflected in the EEG's (low) beta band (13–25 Hz). Beta power shows pronounced decreases in the motor cortex contralateral to the responding side during response preparation and reflects competition between response options in a lateralised fashion[28–30]. While these characteristics of beta power as an index of evidence accumulation have been exploited mainly to study decision making[31], they conceivably also reflect effects of cognitive control processes. We suggest that beta power lateralisation (BPL) over central EEG electrodes can be used as a neuronal readout of the decision-making process and post-error effects. We argue that BPL peak lateralisation can serve as a proxy for the response threshold and that its slope represents the speed of net evidence accumulation. According to this, higher peak BPL would indicate increased response thresholds on post-error trials. Post-error BPL slope should be less steep when the error-induced orienting reflex results in attentional disengagement. In contrast, adaptive suppression of evidence accumulation from distractors should be reflected in a reduced distractor-induced deviation of BPL towards the incorrect side on post-error trials. The dissociable predictions made by adaptive and maladaptive (orienting) accounts for the DDM and BPL results are summarized in Table 1.

To contrast adaptive and maladaptive accounts of post-error adjustments, we make use of behavioural and EEG data from a large sample of 863 young healthy participants, which we combine with computational modelling. Participants performed a flanker task well suited to study cognitive control processes[32,33] (Fig. 1a). The task included 50% congruent trials (target and flanker arrows indicate the same direction) and 50% incongruent trials (distracting flanker arrows induce a response tendency opposite to the target-induced correct response). Additionally, we included manipulations of flanker-target distance (close, far) and

**Table 1 Predictions of different accounts of post-error adjustments at a mechanistic level, in the framework of the multistage drift-diffusion model, and for beta power lateralisation**

| Account | Response threshold | Selective attention or weighting of evidence from different perceptual sources | | | Decision parameters in the multistage drift-diffusion model | | | | beta power lateralisation (BPL) | | |
|---|---|---|---|---|---|---|---|---|---|---|---|
| | | Flanker input | Target input | Weighting flanker vs. target | Boundary | Drift rate | flanker weight | Non-decision time | Peak amplitude | Early flanker-induced BPL to wrong side | Early BPL slope |
| Orienting | +[a] | – | – | **x** | +[a] | – | **x** | +[a] | +[a] | x or – | – |
| Adaptive | +[b] | –[c] | +[c] | **–[b]** | + | **x or +** | **–[b]** | x | +[b] | **–[b]** | **x or +** |
| Findings | + | – | **x** | – | + | – | – | x | + | – | **x** |

+ = increased in post-error relative to post-correct trials; − = decreased in post-error relative to post-correct trials; x = no change; bold: dissociations in predictions of the two accounts
[a]The orienting account could explain post-error slowing by two mechanisms that could replace each other: increased response threshold (i.e., reduced corticospinal excitability, reflected in increased boundary in the DDM and higher peak amplitudes of BPL) or prolonged non-decision time
[b]The adaptive account could explain post-error increases in accuracy by two complementary and mutually non-exclusive mechanisms: increased response threshold enabling longer evidence accumulation and/or suppression of evidence from flanker input relative to evidence from target
[c]The account does not distinguish between enhanced target processing or suppressed flanker processing

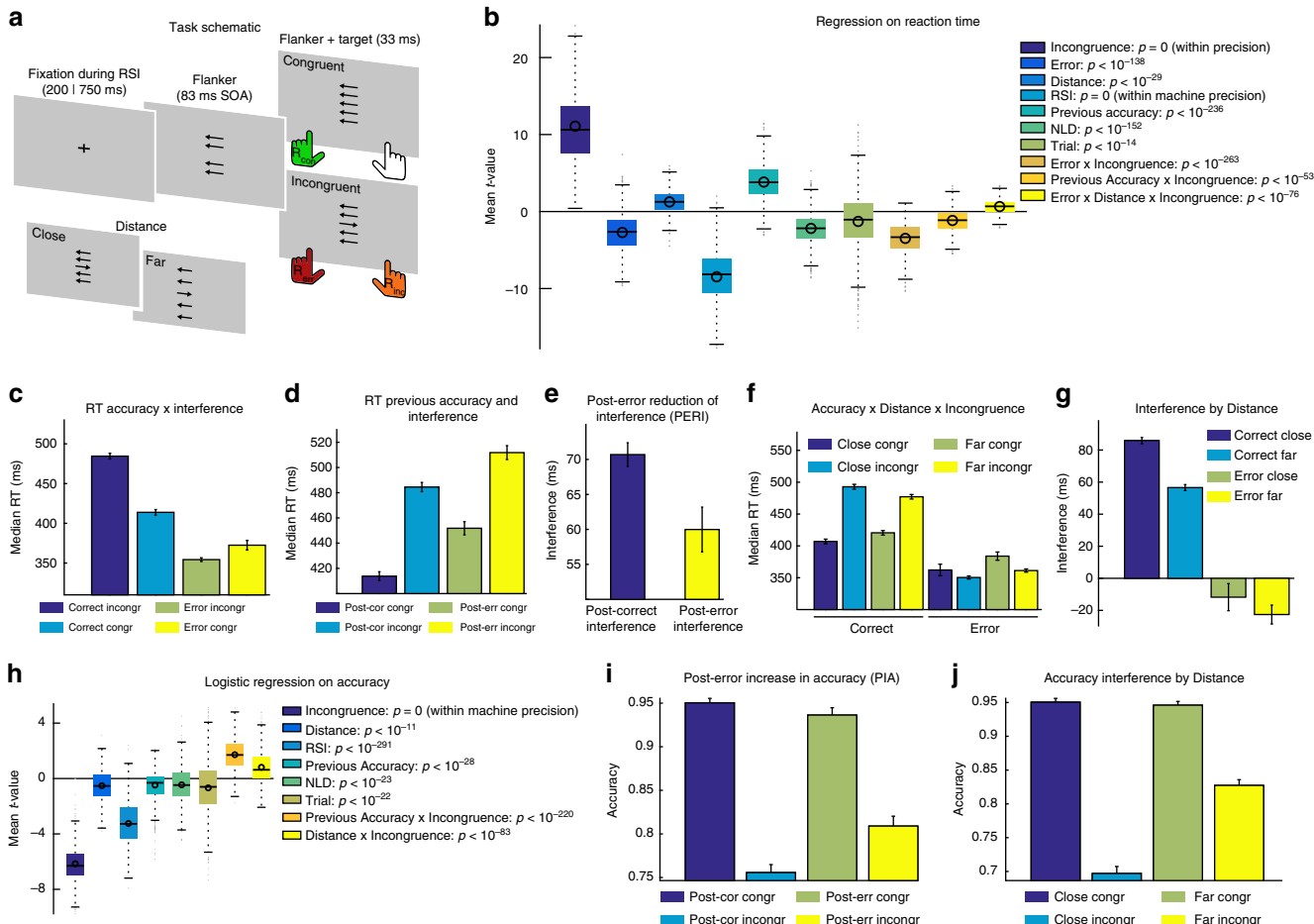

**Fig. 1** Task and behaviour. **a** In the flanker task either congruent or incongruent flankers (four surrounding arrows) in close proximity or further away from the central target (inlet) are presented on each trial. Congruent stimuli usually yield faster responses (green) compared to incongruent stimuli, which lead to delayed responses (orange) or induce errors (red). **b** Multiple single-trial regressions on RT and logistic regression on accuracy (**h**) were used to evaluate participant's behaviour in the task while controlling for confounds and the interdependence of effects. We confirm that stimulus incongruence increases RT and decreases accuracy, while interacting with the distance between flankers and target (**f**, **j**) for both speed and accuracy. Additionally, RT changes depending on stimulus interference interact with accuracy: on error trials, incongruent stimuli have lower RT than congruent stimuli (**c**). Interference effects are furthermore reduced following errors with regard to both reaction speed and accuracy (**d**, **e**, **i**). **b**, **h** display average within participant $t$-values, $p$-values are derived from $t$-tests of individual regression weights against zero. Interference effects are calculated by subtracting RT (**e**, **g**) in the congruent from the incongruent condition. RT is calculated as the mean of within-participants median RTs per condition, RSI = response stimulus interval, NLD = distance since last break. For **b**, **h**, boxes = interquartile range (IQR), o = median, - = mean, whiskers = 1.5 × IQR, grey dots = outlier. Error bars represent 99.9% CI; **d**, **e** include correct trials only

the interval between response and next stimulus, thereby varying factors known to influence interfering effects of distractors and behavioural post-error adjustments, respectively[16,34]. Behavioural analyses confirm that participants in this task display PIA, PERI, and PES. We demonstrate that a multistage DDM that weighs evidence from flankers and targets differently, captures the behavioural hallmarks of the flanker task. Next, we show that BPL represents a non-invasive on-line neuronal correlate of response selection enabling inferences on underlying decision-making parameters and their post-error adjustments: BPL time courses are remarkably similar to the DDM simulations. By separately fitting the DDM to post-error and post-correct trials and by comparing BPL for these conditions, we investigate which model-parameters reflect post-error adjustments. We find that error-related changes in cognitive control can partly be explained as consequences of an orienting reflex, but there is clear evidence for additional adaptive changes observable during stimulus processing and response-selection following errors in cortical beta power.

## Results

**Regression models**. To establish that participants followed task instructions and that all error-related effects associated with cognitive control (PERI, PIA, PES) are present, we determined critical factors that influence RT (GLM 1) and accuracy (GLM 2) in the task in two multiple robust regression models using each participant's single-trial RT and accuracy. Within-participant regression weights were tested for significance using two-sided $t$-tests corrected for multiple comparisons on group level. We followed up regression effects by binning the raw data according to significant factors. Results of these analyses are presented in Fig. 1. Behavioral findings were typical for flanker tasks and reflected effects of interference, the interval between last response and next stimulus (RSI), and flanker distance.

**Post-error adjustments**. GLM 1 confirmed PES by displaying a significant main effect for *Previous Accuracy* ($t_{862} = 46.6$, $p < 10^{-236}$, Fig. 1b). Participants responded 37 ± 3 ms (median ± SE) slower after committing an error. Additionally, previous

error commission reduced the degree of interference exerted by distractors confirming PERI (Fig. 1d, e). Following errors, participants displayed increased accuracy on incongruent, but not on congruent trials (Fig. 1i, j, interaction *Previous Accuracy × Incongruence* $t_{862} = 21.8$, $p < 10^{-83}$), which implies that PERI influences both the speed of stimulus processing and its accuracy, seen in the form of PIA.

A possible confound for these effects is that stimulus incongruence causes errors. Therefore, post-error trials are mostly preceded by incongruent trials which themselves cause conflict adaptation effects[9,35,36]. We ensured that all three adaptation effects could not be explained by conflict adaptation via inclusion of the previous trials' congruency into the regression model (Supplementary Note 1).

**An extended DDM captures task effects**. Simple accumulator models integrate sensory evidence into a decision variable that determines choices and RT. Intracranial recordings in monkeys[37,38] in various cortical areas revealed many neurons reflecting this evidence accumulation and triggering a response when a common threshold was reached. A simple variant of these accumulator models is the DDM which assumes that choice options are mutually exclusive. This allows to replace two separate accumulators with one decision variable reflecting the difference between both decision options. On a given trial, the decision can be randomly biased to favour one response (start-point variance, *sz*). The height of the boundary parameter (*a*) determines how much evidence accumulation is required to cross the boundary and trigger a response. Finally, visual processing and motor execution times are captured by the non-decision time ($T_{er}$) parameter, which in our model can vary from trial to trial (*st*).

Usually, DDMs assume a constant speed of evidence accumulation within each trial, which is governed by a drift rate parameter (*v*, Fig. 2a) and trialwise variance (*sv*). However, in our flanker paradigm, evidence can first point into one direction and thereafter reverse. To reflect this, we used a multi-stage DDM. In our model, evidence accumulation followed the flankers' direction during the time they were displayed on screen alone (83 ms). After target onset, evidence accumulation was driven by the direction of the target.

We fit three models to the distribution of RTs for each participant using quantile maximum-likelihood estimation[39] and differential evolution algorithms[40]. The first model was a standard DDM using the same evidence accumulation rate for distractor and target stimuli. The second DDM included the possibility to relatively downweigh (suppress) distractor evidence (parameter *f*). The third model additionally allowed trial-by-trial variance in distractor weighting (*sf*). Model comparison (Fig. 2b) suggested that the last model provided the best fit to the data, indicating that participants suppress distractor information, yet this varies between trials. Suppression of distractors was confirmed by the value of *f*, which was significantly lower than 1, where 1 would reflect equal processing of distractor and target information (mean *f* = 0.44 ± 0.03 (99.9% CI), $t_{862} = 64.3$, $p = 0$ within precision). Overall, the model provided a good fit to the data and matched participants´ RTs on congruent, incongruent and error trials as well as accuracies in congruent and incongruent trials (Fig. 2c–f). Parameter recovery analysis confirmed that our fit method reliably identified model parameters (Supplementary Figure 3). We used this model to provide a quantitative analysis of post-error effects and make predictions on how such a decision variable should behave when it is analysed for task effects.

**Association between model parameters and behaviour**. We then tested which model parameters were associated most strongly across participants with RT, accuracy, interference and the ratio of errors in congruent relative to incongruent trials (Fig. 2g–m, GLM 3, Methods). This revealed that RT was mainly reflected by the drift rate (*v*), accuracy by boundary (*a*) and distractor weighting (strength *f* and variance *sf*). The distractor weighting parameters additionally covaried with the magnitude of the interference effect (Fig. 2l) as well as with the ratio of incongruent to congruent errors (Fig. 2m). Notably, the model also predicted errors on congruent trials (Fig. 2f), although these were removed for fitting (see Supplementary Methods).

We then investigated which variance parameters on a single-trial basis affect the models' accuracy and RT using a similar regression approach as in the behavioural analysis (GLM 4, Methods): Both accuracy ($p < 10^{-10}$) and RT ($p < 10^{-33}$) were significantly reduced when, due to variance in start-points (*sz*), a trial was biased towards the later-on-selected response (Fig. 3). Additionally, the variance of trial-wise suppression of distractor information (*sf*) was strongly associated with accuracy ($p < 10^{-52}$) and less with RT ($p < 10^{-9}$). This analysis identifies that especially variance in start-points and flanker weighting are factors driving speed and accuracy of individual trials in the model.

**Post-error adaptations in the DDM**. Next, we fit the same DDM to post-error and post-correct trials separately. We excluded 15 participants with less than 30 valid post-error trials. To facilitate convergence, we fixed the variance parameters of the DDM (*sv*, *st*, *sz*, *sf*) to the group mean. Note that variance parameters were still in the model; only their value did not change between participants. The DDM captured RT and accuracy well in both post-error and post-correct trials (Fig. 4a–e).

We then compared differences in parameter values for drift rate, boundary, non-decision time and flanker weighting between both models. To this end, we used multiple logistic regression of parameter values onto which trials the model was fit to according to Eq. 5 (Methods). Positive regression coefficients indicate parameter increases following errors. We found that drift rate was decreased ($t_{1691} = -15.3$, $p < 10^{-51}$; Fig. 4f), boundaries increased ($t_{1691} = 17.5$, $p < 10^{-66}$), and non-decision time unchanged following errors ($t_{1691} = -1.7$, $p = 0.1$). Additionally, flankers were more strongly suppressed ($t_{1691} = -5.3$, $p < 10^{-6}$) in post-error trials. This suggests that slower evidence accumulation began at the same time and was less susceptible to distractors following errors.

**Beta power lateralisation**. We confirmed that motor preparatory beta signals are effector-specific: beta power decreased more at centro-parietal electrodes contralateral to the initiated response (C3/4 and CP3/4; Fig. 5a). Subtracting the power of the contralateral from the ipsilateral hemisphere results in BPL: More negative values of BPL reflect beta decreases in favour of the response that will be chosen. The lateralisation peaked at the time of movement execution (Fig. 5b, mean peak latency relative to button press = 3.1 ms ± 2.8). Flanker-locked analysis was used to investigate stimulus processing, and response-locked analysis to assess BPL changes associated with the response—which may reflect boundary changes[41]. The baseline-free BPL intrinsically and continuously reflects differences between two measurement sites, and thus unbiasedly extends the temporal range of analyses into periods often used as baselines.

**BPL reflects characteristics of the DDM decision variable**. We compared BPL during stimulus processing against predictions of the DDM. The predictions were derived by simulating 5000

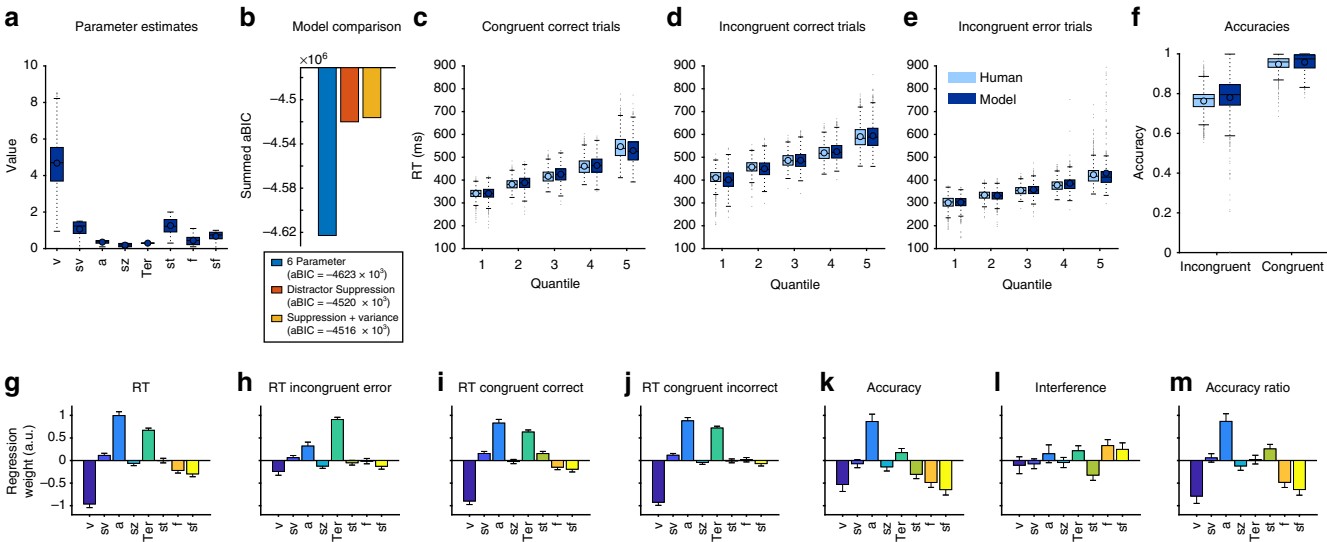

**Fig. 2** Drift-diffusion model fits and parameters. The drift-diffusion model (DDM) included eight free parameters (**a**). Apart from the standard-parameters reflecting drift rate (*v*), boundary (*a*), and non-decision time (*Ter*), we included variance parameters that determined how much standard-parameters varied from trial-to-trial (*sv*, *sz*, *st*, cf. Supplementary Fig. 1). Model comparisons (**b**), revealed that two additional free parameters for distractor weighting (*f*, red) and it's trialwise variance (*sf*, yellow) increased model fit measured as approximate BIC (see Supplementary Figure 2 for results of worse fitting models). *f* varies with variance *sf* over trials and modulates flanker processing by scaling the drift rate during distractor presentation. (**c–e**) shows quantile fits of the model (dark blue) against human RT data and (**f**) shows model and human accuracy. In all conditions (congruent & incongruent correct as well as incongruent error), the model captures the RT data in each quantile, suggesting a good fit to the data. Plot conventions as in Fig. 1b. Note that we removed congruent errors from the analysis as these were very rare (<2.5% of trials). **g–m** Displays the relationship between model parameters and behaviour across subjects. Displayed are regression coefficients and 99.9% confidence intervals. **g** Faster participants were fit by higher drift rates, lower decision boundaries, and lower non-decision times. **h–j** RT on error trials **h** was most strongly dependent on the non-decision time because errors are usually very fast, whereas drift rates are more closely associated with RT on correct congruent (**i**) and incongruent (**j**) trials. **k** Accuracy was most strongly reflected in the height of the boundary (*a*) but additionally dependent on how much distractors were processed (*f*) and how variable their suppression from trial-to-trial was (*sf*). Higher variance reduces accuracy, because in more trials distractors are likely to not be suppressed. **l** Interference and the ratio of errors between incongruent and congruent trials (**m**) additionally covaried with distractor weighting and its variance (*f*, *sf*): the less flankers were suppressed in the model, the higher was the interference and the more incongruent compared to congruent errors a participant made

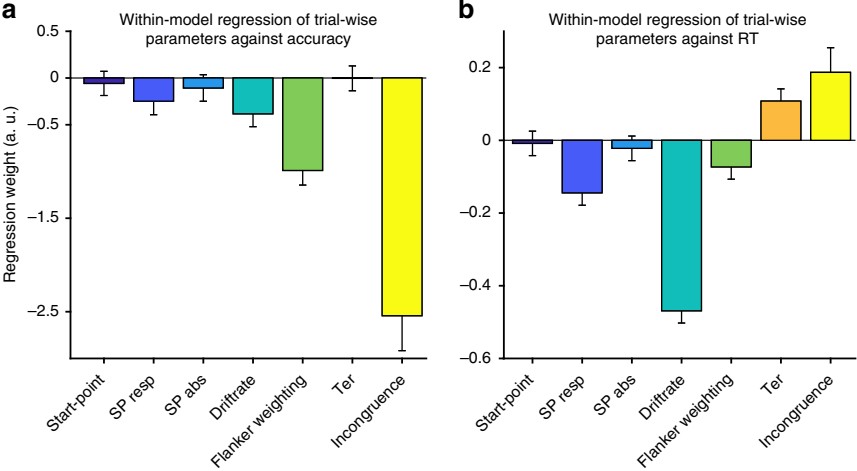

**Fig. 3** How model parameters influence accuracy and RT. Displayed are regression coefficients of a single-trial regression analysis within the DDM using trial-wise variance parameters. The goal of this analysis was to delineate which parameters have strong impact on whether a trial will be correct (**a**) or fast (**b**). Akin to the participant data (Fig. 1), we used logistic and multivariate regression to investigate accuracy and RT, respectively. In both models, a trial's incongruence had a strong impact (yellow, compare to Fig. 1b, h). Variance in the start-point of the decision process of each trial did not overall influence accuracy and RT. However, how much the start-point was biased towards the response the model later on selected, significantly decreased accuracy and RT (factor *SP Resp*). This effect was less pronounced for the absolute bias per trial (factor *SP Abs*). Trialwise faster drift rates significantly decreased RT and much less so accuracy. When flankers were more strongly suppressed (*f* is lower), accuracy increased and RT slightly decreased. Non-decision time ($T_{er}$) did not influence accuracy, but longer $T_{er}$ on each single trial led to increased RT. Error bars reflect 99.9% CI, plotted are regression weights for one DDM that employed the group mean parameters for all post-correct trials and simulation of 5.000 single trials

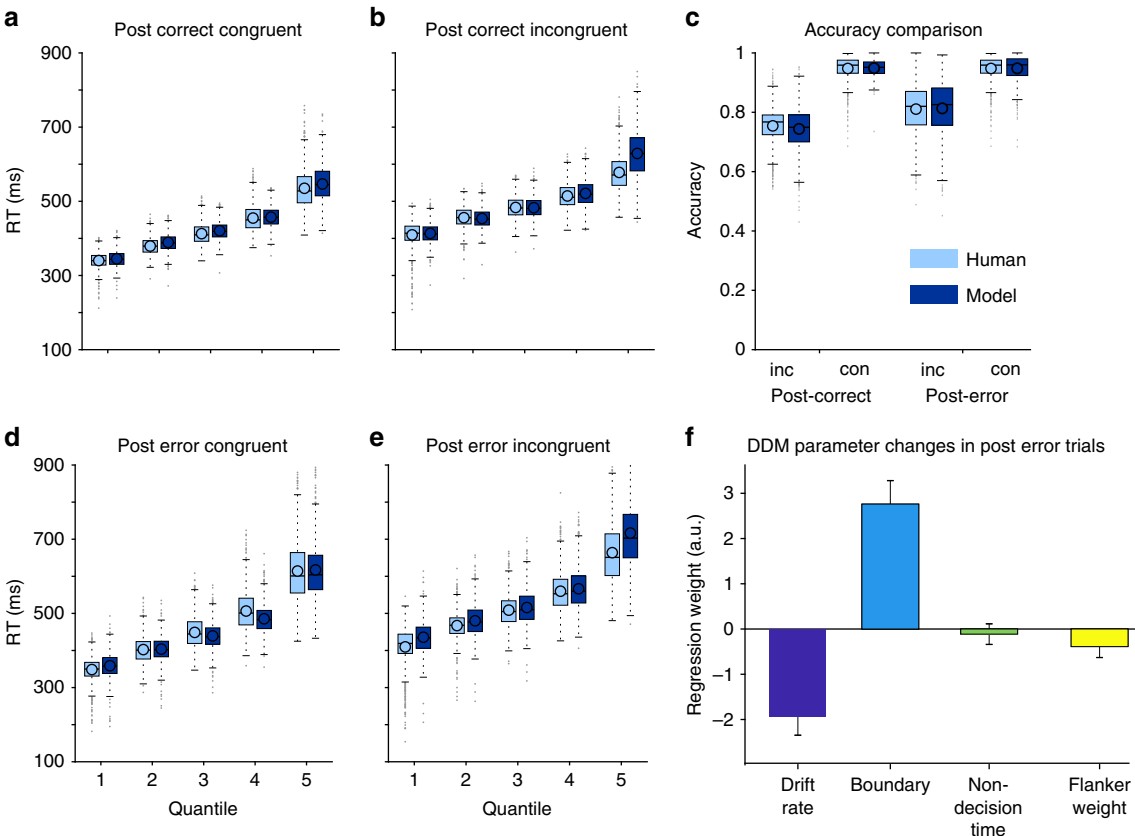

**Fig. 4** Model changes following errors. When we fit the model separately to all trials following correct responses (**a**, **b**) and all trials following errors (**d**, **e**), we again observed that the model captured RT distributions in all quantiles of the data despite partly limited trials in the post-error condition. By comparing (**a**) against (**d**) and (**b**) against (**e**) it can also be seen, that the model captured PES. Furthermore, the model displayed significant accuracy increases especially in incongruent trials (**c**) post-error ($t$-test $t_{848} = 17.5$, $p < 10^{-56}$). **f** Model parameter values were compared using logistic regression onto *TrialType* (post-error, post-correct) according to Eq. 5. Plotted are regression weights and standard errors for individual factor contributions to the difference between models fitted to post-error and post-correct trials and positive values indicate parameter increases following errors. The largest change in parameter values was observed for decision boundaries, which we found to be increased when the model was fit to post-error trials. Additionally, drift rates were decreased, yet the non-decision time was unchanged. Crucially, the degree to which distractors influenced the decision variable (multiple linear regression $t_{1691} = -5.3$, $p = 1.24 \times 10^{-7}$) was decreased following errors, which was seen over and above decreased drift rates or increased boundaries. Plot conventions for a-e as in Fig. 3, error-bars in (**f**) reflect 99.9% CI

individual trials using the mean of the best fitting individual subjects' parameters. For comparison with the EEG data, we extended the model's trial duration such that a pre-stimulus baseline matched the recorded EEG signal. The baseline in the model was influenced by start-point variance. We additionally assumed that the diffusion in the model returned to zero after a response (see Supplementary Methods). As in the EEG analysis, trials in the DDM were grouped by stimulus congruence, accuracy and RT (via median split). Note that the DDM was not fit to the neural signal.

In both signals, distractors began to influence the decision at around 280 ms following onset of the distractor stimuli, which coincides with the fitted non-decision time parameter (Fig. 5c, e). A pronounced shift towards the incorrect response was seen on incongruent but not congruent correct trials, peaking at 350 ms. This distractor-related deflection in BPL positively correlated with the behavioural interference effect on RT (Fig. 5d), confirming that this signal explained interindividual variance in conflict processing. Furthermore, we found that BPL reflects response conflict modulation by distractors' distance from the target (Supplementary Figure 6).

Even before stimulus onset, the DDM predicted differences in the decision variable and all these predictions were matched by

BPL. Incorrect responses in the DDM are more strongly lateralised towards the boundary of the incorrect response (Fig. 5g) and the same is seen in BPL even before stimulus onset (Fig. 5f). Moreover, error RTs in the model and human data precisely coincide with the peak of distractor processing on incongruent correct trials (plotted in blue), possibly indicating similar mechanisms inducing errors. Like errors, fast correct responses in the DDM are more strongly lateralised to the boundary during the baseline and this is again seen in BPL (Fig. 5h, i). A baseline difference is present for incongruent trials both in the model and in BPL (Fig. 5c, e), which can be interpreted as a result of the correct-only trial selection: Since incongruent trials with pre-existing (random) response tendencies towards the incorrect response (leading to errors) were discarded, this results in visible disproportional pre-activations. Response-locked data are shown in Supplementary Figure 6 for these comparisons. Furthermore, analysing BPL as well as the separate contributions of contralateral and ipsilateral beta-power onto upcoming speed and accuracy revealed that only the relative degree of BP favouring one or the other response was predictive of both choice accuracy and speed even before stimulus onset (Supplementary Figure 5). Therefore, BPL remarkably resembles the influence of *start-point variance* in the DDM which cannot be

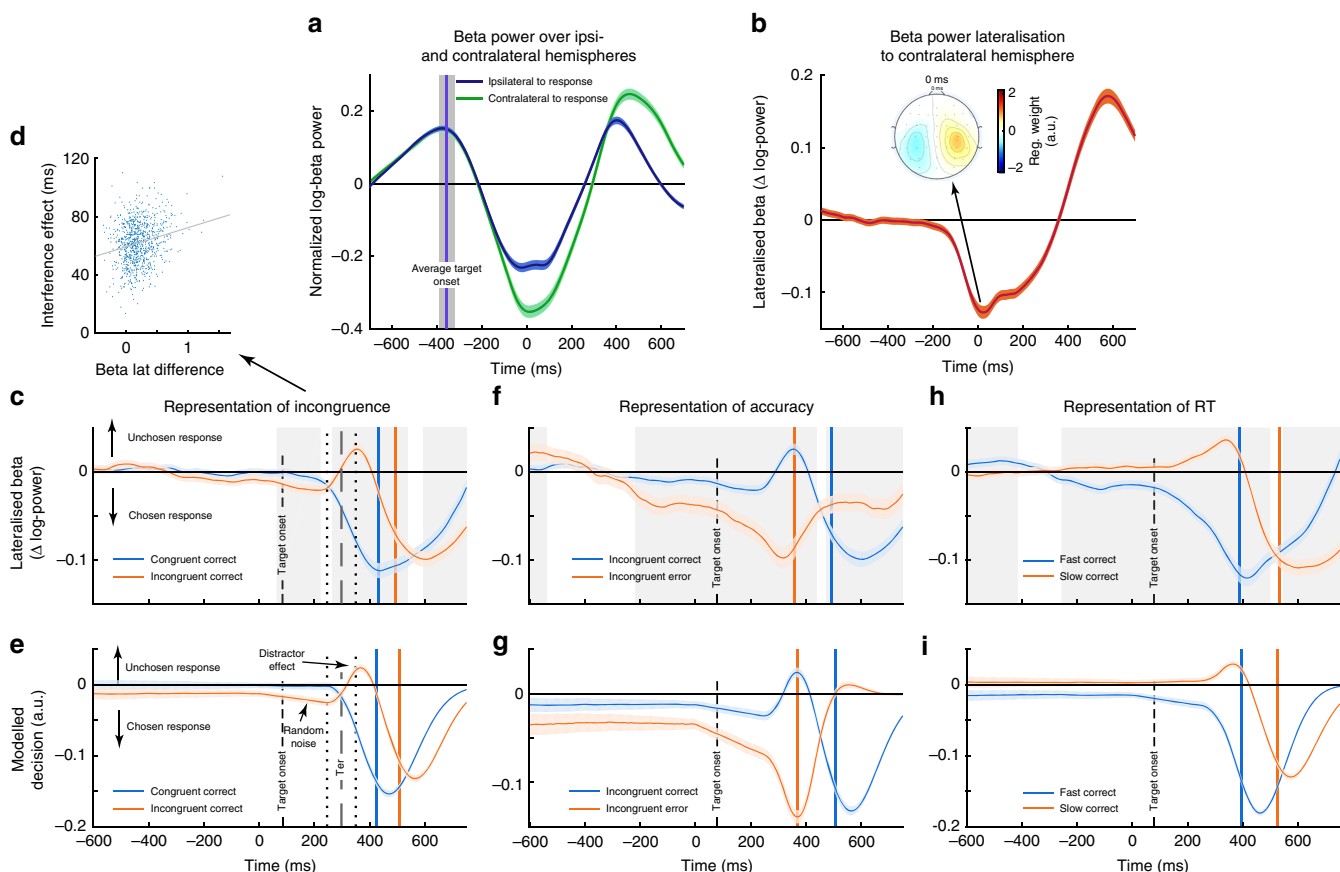

**Fig. 5** Stimulus related beta power and simulated decision variable. **a** The BPL is composed of the activity-difference recorded over contralateral (green) and ipsilateral motor cortices (blue). A statistically significant lateralisation towards the contralateral motor cortex is seen around 200 ms prior to response execution (**b**). The inset shows a regression weight map of the scalp topography of beta power when regressed against the side of motor execution according to Eq. 6 (see Methods and Supplementary Figure 4 for details). **c**, **e** Comparison of distractor effects on BPL (**c**) and modelled decision signal in the DDM during stimulus processing (**e**). Distractor effects were seen at similar times on incongruent (red) compared to congruent trials (blue): distractors pushed BPL and decision variable away from the correct response. **d** Across participants, the degree of distractor representation in BPL (mean between 340–420 ms) positively correlated ($r = .20$, $p < 10^{-16}$) with the interference effect on RT. Notably, many features of the time-course and dynamics of BPL were closely predicted by the diffusion variable of the DDM. Non-decision time ($T_{er}$) reflects stimulus processing without influence of stimulus information and the onset of BPL matches $T_{er}$ plus its variance (dots surrounding vertical dashed line in **c** and **e**). The similarities extend to pre-stimulus effects. Here, both signals are lateralised towards the chosen response which is caused by single-trial start-point variance in the DDM affecting accuracy and RT. Strong pre-stimulus bias leads to errors (**g**) but also quick (based on median splitting) responses (**i**) in the model. The same patterns are evident in the BPL plots (**c**, **f**, **h**), suggesting that BPL influences speed and accuracy of upcoming responses. In the DDM, this is caused by the causal relationship between start-point variance and accuracy as well as RT. The striking similarity between diffusion signal and BPL hints at similar relationships for the neural data. Note that the pre-stimulus bias is absent when trials are plotted without mapping to the executed response. Shades represent 99.9% CI, the vertical blue and orange lines in **c**–**i** reflect mean RT per condition. Grey backgrounds in (**c**, **f**, **h**) indicate significant time-points after Bonferroni correction. For details about the simulation see Methods and Supplementary Figure 1

seen when investigating the signal over either hemisphere alone. Overall, these striking similarities between modelled signal and BPL suggest that BPL can be used as a marker of decision formation to investigate more complex effects of cognitive control related to error processing.

**Error-induced beta power changes**. Fitting the DDM to post-error and post-correct trials separately revealed an increased boundary parameter following errors. Because threshold increases in the DDM result in prolonged evidence integration, this mediates both slower and more accurate responses, an effect predicted by both the adaptive and the orienting account (Table 1). Compatible with the DDM and increased decision boundaries, post-error BPL was increased at response execution (Fig. 6a–h). We confirmed this finding on a single-trial level by regressing task-related behavioural factors onto measured individual single-trial

BPL at response execution (button press ± 12 ms). This allows to account for other task factors (Eq. 7, Methods) that may have additional influence on BPL, and to test if effects are independent from RT. The regression confirmed that BPL at response execution was increased on post-error trials (main effect *Previous Accuracy* $t_{862} = -12.2$, $p < 10^{-31}$; Fig. 7). Because BPL reflects the difference between contralateral and ipsilateral hemispheres, changes can result from effects on either hemisphere. Therefore, we performed the same regression analysis separately for both ipsilateral and contralateral electrodes. We found that beta power over both hemispheres was increased following errors (Fig. 7g). The BPL difference was due to more pronounced increases over the ipsilateral hemisphere ($t_{862} = 13.9$, $p < 10^{-39}$, contralateral: $t_{862} = 6.2$, $p < 10^{-8}$). In incongruent trials, beta was reduced over the ipsilateral cortex ($t_{862} = -17.5$, $p < 10^{-57}$, contralateral: $t_{862} = -1.1$, $p = 1$) as well, which can be interpreted as enhanced motoric readiness caused by the flanker stimuli. Interestingly, this

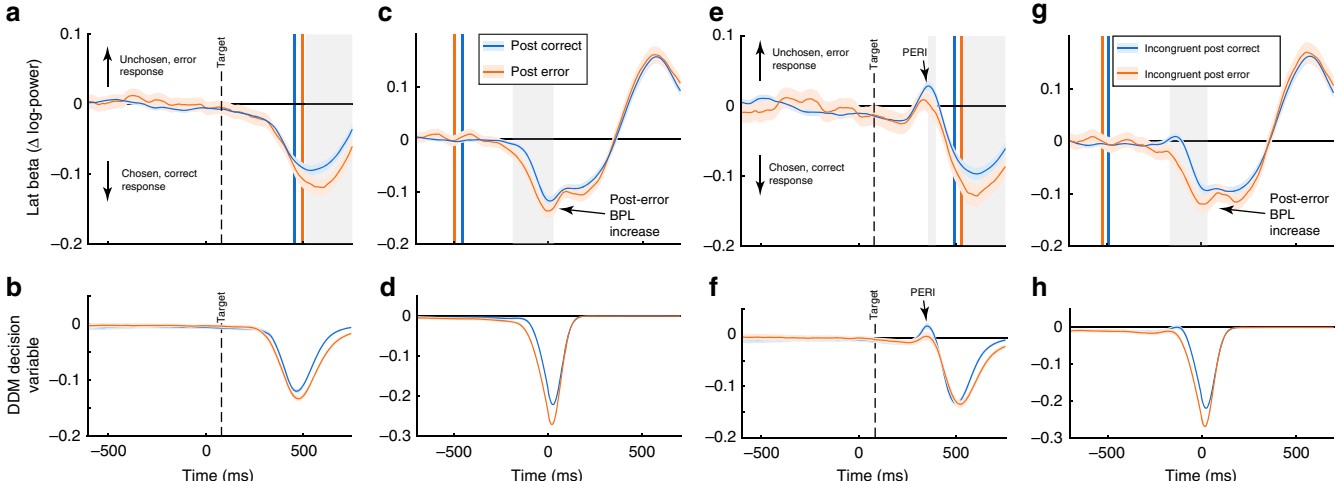

**Fig. 6** Post-error adaptations in beta power and simulated choices. Plotted are BPL time-courses following error and correct responses for all trials (**a–d**) and incongruent trials specifically (**e–h**). In the lower row the simulated decision signal derived from the DDMs fit to post-correct and post-error trials is plotted split up by the same factors . We found evidence for increased BPL at response execution (**c**), which in the DDM is caused by boundary increases following errors (**b, d**). Additionally, post-error reduction of interference (PERI) is seen in the beta signal (**e, g**) and predicted by the DDM decision variable as well (**f, h**). We additionally confirmed that not only post-error adaptations influence distractor processing, but also that the BPL signal is in general sensitive to the degree of response conflict induced per trial by comparing close and far distractor trials (Supplementary Figure 5). Shade = 99% CI, grey background = significant time points between conditions after Bonferroni correction. Vertical coloured lines = flanker/response onset per category as in Fig. 5

incorrect activation is reduced after errors, speaking for more efficient flanker suppression which is seen as increased BPL at response ($t_{862} = 12.6$, $p < 10^{-32}$). These differential findings suggest that BPL and general beta power reflect distinct properties of the decision process.

Additionally, we found that higher RT per trial was associated with decreased BPL at response ($t_{862} = 13.4$, $p < 10^{-35}$; Fig. 7a, e, h). This effect was stronger than the effect on either hemisphere alone (cf. Supplementary Figure 7). Reduced BPL is compatible with the notion of dynamic decision boundaries[42], which adaptively decrease under time-pressure or urgency. Thus, threshold modulations appear to be reflected especially in BPL. Thresholds are furthermore compatible with a speed-accuracy trade-off induced by two additional factors, which were seen over-and-above the effect of RT. First, following a longer RSI we found significantly faster and less accurate choices (Fig. 1b, h) which were associated with reduced BPL ($t_{862} = 23.8$, $p < 10^{-95}$; Fig. 7a, c). Furthermore, with increasing duration since the last break between blocks in the task (factor $NLD$ = negative log-distance from break), responses were faster and less accurate (Fig. 1b, h), again accompanied by decreased BPL (Fig. 7a, $t_{862} = 4.5$, $p < 10^{-4}$). The distance between flanking and target arrows and trial number in the task did not significantly change BPL ($ps > 0.47$). Therefore, response-related increases in BPL are in accordance with trial-by-trial increases in decision thresholds resulting in increased accuracy at the cost of speed of the decision process.

**Dynamics of response selection after errors**. Although the previous findings appear to favour the adaptive error processing account, the contribution of a (weak) orienting response could still be compatible with the data. Therefore, we searched for more neural evidence for increased attentional control that could dissociate both accounts. On incongruent post-error trials we found a beta band effect that reflected PERI (Fig. 6e, f). The influence of incongruent flankers on BPL was significantly reduced compared to post-correct trials in accordance with increased flanker

suppression in the DDM. This indicates suppression of distractor processing consistent with the adaptive error processing account. Alternatively, early disengagement of selective attention, as predicted by the orienting account, could act particularly on processing of the flankers preceding target onset. However, we found no change in the non-decision time parameter in the DDM analysis following errors, speaking against a general delay of choice formation. Still, we additionally tested if BPL suggested delayed choice formation following errors.

**BPL slope analysis**. Whereas the adaptive error processing account predicts a selective suppression of flanker processing, the orienting account predicts attentional disengagement resulting in general slowing of evidence accumulation that might be most prominent earlier (during flanker processing). We quantified the steepness of the slope of BPL following correct and error responses locked to flanker onset using a simple regression line fit through an 80 ms interval surrounding the average individual participant data points with a step-size of 10 data points. We aimed at congruent trials to test if processing of information was delayed in general following errors and rule out confounding PERI effects. We tested at what time BPL slopes in congruent and incongruent trials first deviated significantly from zero (following Bonferroni correction over time). For both trial types this was at 220 ms (straight line in Fig. 8a). Note that this closely coincides with the fitted non-decision time that indicated decision processing beginning 230 ms after flanker onset (group mean $T_{er} = 290$ ms, $st = 120$ ms, earliest onset = $T_{er} - st \times 0.5 = 230$ ms). We furthermore found that BPL slopes are sensitive to changes in distractor processing induced by visual proximity and reflect post-error reductions in distractor processing (Fig. 8b, c). These analyses support the notion that BPL slope reflects the net speed of evidence accumulation. Contrary to the predictions by the orienting account, on congruent post error trials, BPL slope steepness was not significantly reduced (Fig. 8d). However, later BPL slopes were even steeper following errors, which is compatible with

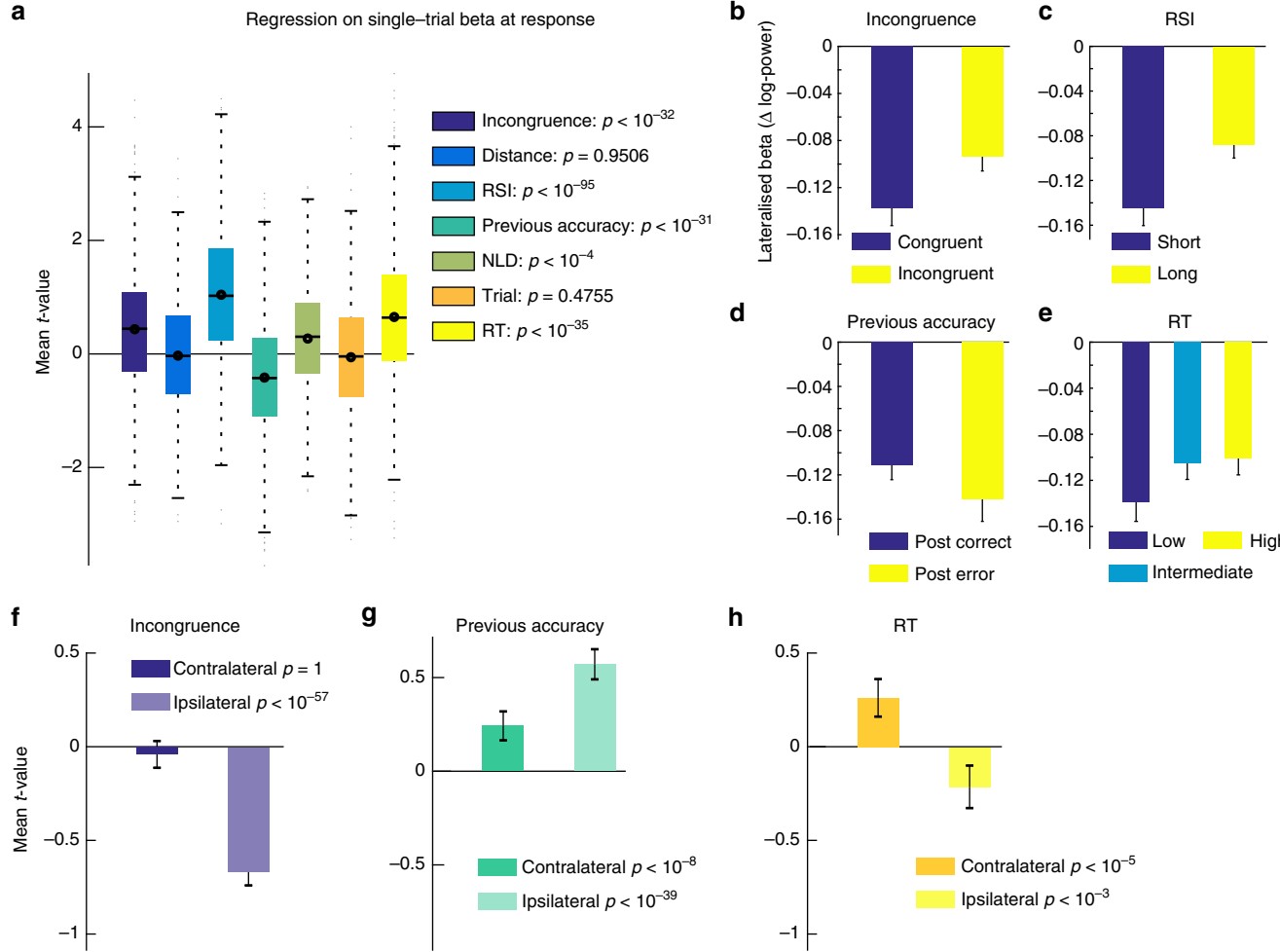

**Fig. 7** Single-trial regression on beta at response. **a** Results of a single-trial regression analysis on BPL at response (mean ± 12 ms surrounding button press), comparing relevant task factors. This analysis was performed to rule out that the results demonstrated for RT and post-error effects result from averaging over trials. **b–d** Reflect raw values splits for *Incongruence* (congruent vs incongruent), *RSI* (short vs fast) and *Previous Accuracy* (correct vs error). **f–h** Display results of separate regression analyses split by ipsilateral (hemisphere that does not induce the choice on the current trial) and contralateral hemispheres. Following errors, BPL is larger (more negative) which is caused by a stronger increase in BP over the ipsilateral hemisphere, suggesting that following errors, the unchosen response is more effectively suppressed. Note that in general (Fig. 5a), beta power strongly decreases before responses are executed. When responses are slow, BPL is decreased. This is partly due to less decreases over the contralateral hemisphere, but the effect is much stronger when the relative signal per trial is investigated than when either hemisphere on its own is analysed (**h**). Furthermore, incongruent trials cause reduced BPL. This latter result is induced by changes over the ipsilateral hemisphere (**f**) where BP more strongly reduced on incongruent trials, which could reflect the preactivation of the competing response tendency. Error-bars reflect 99.9% CI, **a** displays mean within participants *t*-values, and statistics are results of *t*-tests of individual within-subject regressions against zero. RT in (**e**) is trichotomised. See Supplementary Figure 5 for a similar analysis on pre-stimulus BP. NLD reflects the distance from the last break in the task, and trial number the general time on task

focussing of selective attention to the target arrow. As a net effect, accumulating evidence for flankers was reduced and counteracted faster by the sensory input from the target, which favours the adaptive account of post-error adaptations.

## Discussion

Using computational modelling with a multistage DDM and the analysis of a putative neuronal readout of decision making (BPL), we investigated whether post-error adjustments in a flanker task can be explained by adaptive cognitive control mechanisms or by (potentially maladaptive) concomitants of an orienting reflex induced by the error. Specifically, we aimed at elucidating the mechanisms underlying the typical behavioural adjustments found in interference tasks, namely PES, PIA, and PERI. The multistage DDM captured the main characteristics of the flanker task, in particular RT distributions for correct congruent, correct

incongruent, and incorrect incongruent trials. This was achieved by introducing a weighting-factor determining the contribution of flanker-information to evidence accumulation. This factor balances fast errors induced by the distractors against attentional regulation.

We then compared DDM time courses predicted for various conditions of the flanker task, such as correct/incorrect, fast/slow or congruent/incongruent trials to BPL We found that BPL and DDM predictions were strikingly similar to each other. We therefore argue that BPL, even in highly speeded RT tasks in which one of both hands executes the response, can be used as a baseline-free measure of the state of evidence accumulation that is sensitive to interference effects both qualitatively and quantitatively. The topography of the lateralised beta power reduction before and around a motor response was in accordance with an origin in the motor cortex. Although the spatial resolution of EEG does not provide means to safely assume the origin of the beta

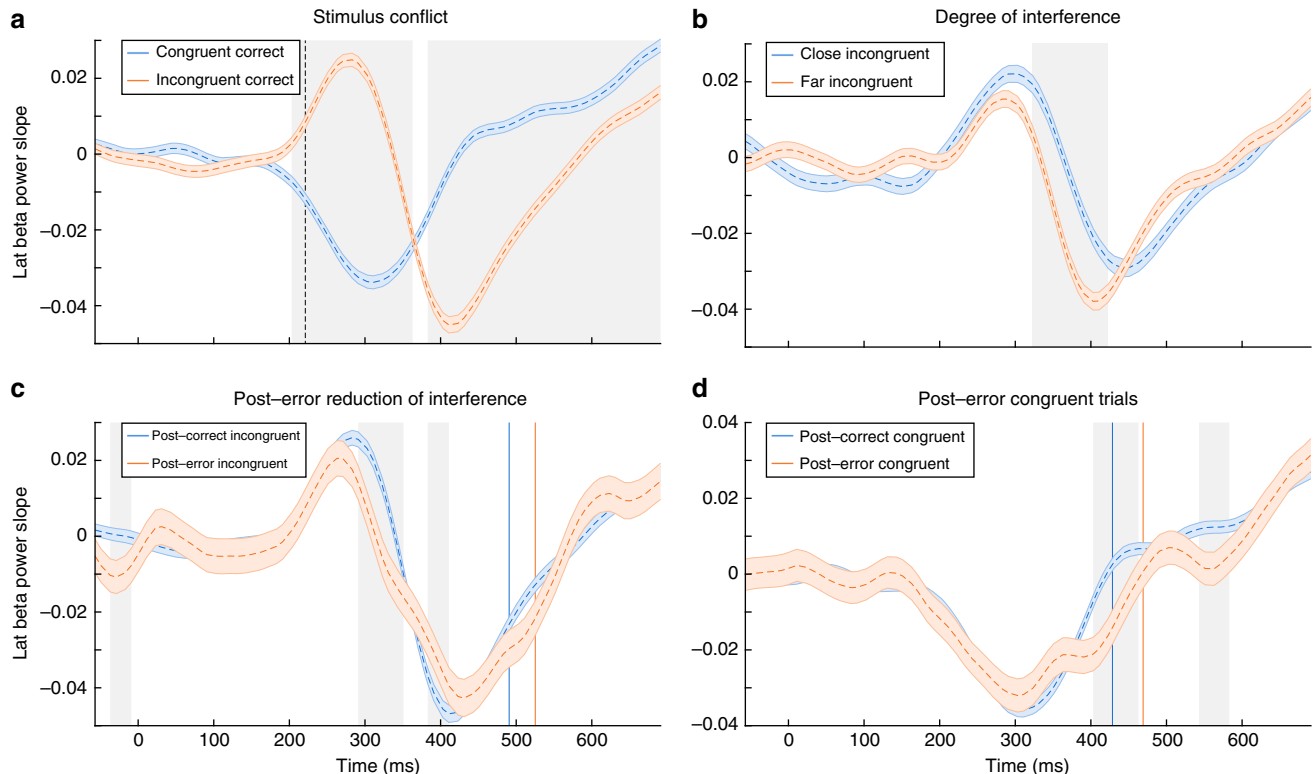

**Fig. 8** Beta slope dissociates specific from general adaptations. BPL slope indexes the steepness of the BPL signal analysed via single-trial fitting of individual periods in the BPL signal surrounding every data point (80 ms). The slope can be seen as the rate of change in BPL, thus reflecting how quickly beta power lateralises to one hemisphere. **a** Displays how BPL slopes differ between congruent (blue) and incongruent (orange) trials. The dashed vertical line reflects when BPL slopes start to first significantly deviate from zero, which coincides with the non-decision time parameter in the DDM. **b** Displays that the degree of interference induced by visual proximity of the distractors affects the duration of BPL slopes that point into the incorrect (distractor-) direction. As a result, when incongruent flankers are close to the target, BPL to the wrong direction is stronger and longer-lasting as compared to far flanker trials (cf. Supplementary Figure 6). Similarly, for incongruent trials the slope of distractor processing is influenced by previous accuracy (**c**), which is consistent with PERI. However, there is no significant difference (even at a very lenient threshold of $p < 0.05$) at any point during early BPL slopes when comparing congruent post-error (orange) to post-correct (blue) trials. This suggests that post-error trials specifically reduce the interference induced by distractors, which is seen in the BPL slope analysis at around 290 to 360 ms. For an additional analysis comparing how slow and fast responses are associated with BPL slope, see Supplementary Figure 8. Statistical comparisons between conditions are calculated via repeated $t$-tests corrected for multiple comparisons (**a**, **b**) or uncorrected (**c**, **d**), and significant time periods are shaded in grey reflecting SE

signal, effector-specific signals have been found in humans in primary and pre-motor cortex extending to the post-central cortex[27,28] and likewise in intracranial recordings in animals[43,44]. BPL direction indicates the prevailing evidence for responding with the hand contralateral to the hemisphere with stronger beta-power decrease. The slope of BPL appears to indicate the net speed of evidence accumulation. BPL peak lateralisation, coinciding with the motor response, may represent a proxy for the motor threshold. However, other factors may influence BPL at response as well. BPL depends on the strength of BP decrease over both hemispheres, which can vary independently. Ipsilateral BP decrease was stronger on incongruent compared to congruent trials and less pronounced on post-error compared to post-correct trials. It is not yet clear whether the relative state of evidence for both choice options, i.e., the BPL, or purely the BP decrease contralateral to the responding hand suffice to trigger a response. Nevertheless, our analyses and the striking similarities to the time course of evidence accumulation predicted by the DDM strongly suggest that BPL reflects many features of the decision variable in very high temporal resolution. These results do not rule out that responses are finally determined by two separate accumulation processes in both hemispheres. However, they suggest that, at least in the context of mutually exclusive decisions, DDM assumptions are neurally validated and BPL

appears to be a sensitive measure of response-threshold modulations.

We used the DDM parameters fitted to the participants' behavioural data and the BPL as a neuronal correlate of the response selection process to test the respective predictions of adaptive and orienting accounts of post-error adjustments (Table 1). Behaviourally, following mistakes, participants were both slower and more accurate, which is in accordance with adaptive changes in cognitive control, but might also be explained by the orienting account, if relative performance adjusted for RT is compared. For post-error trials the DDM showed an increased boundary and the BPL yielded increased peak amplitudes, both indicating increased response thresholds. This explains slower and—resulting from longer evidence accumulation—more accurate responses reflected in PES and PIA[20]. This finding is in accordance with predictions from both accounts and with previous findings[20,22]. Furthermore, we found that distractor influence on response selection was reduced following mistakes. This was demonstrated by a reduced flanker-driven deflection of the BPL in incongruent post-error trials as compared to post-correct trials and was similarly seen in the modelled DDM signal time-courses (Fig. 6). This finding suggests, in line with adaptive error processing accounts, that participants were more efficiently able to suppress distracting information after they made a mistake.

This effect constitutes a rapid adaptation that occurs around 380 ms after onset of the next trial. Because trials here were separated by only up to 700 ms, in less than a second, stimulus processing changes and is focussed towards relevant positions following commission of an error. However, under certain circumstances, an orienting reflex might yield a similar reduction of the incongruence-effect on BPL: Disengagement of attention[10] could generally reduce perceptual processing of stimuli which should result in slower evidence accumulation. Indeed previous studies found evidence for a reduced signal-to-noise ratio in the perceptual signal[20] and a reduced visual P1[19] after errors in perceptual decision making tasks. Thus, a post-error orienting reflex should be associated with slower evidence accumulation which, in turn, could reduce the distracting effect of flankers on BPL. This seemingly specific effect on flankers would be particularly strong, if the effects of the orienting reflex start early and decay quickly as has been suggested previously[16,21,45], because flankers preceded targets. In contrast, adaptive accounts suggest post-error increases in selective attention[15,46] or inhibition of conflicting response tendencies[14] which could be reflected in enhanced drift rates and/or relative suppression of flanker-driven evidence compared to target-driven evidence. In sum, DDM and BPL data provide more evidence for adaptive post-error control, up-regulating selective attention and selectively suppressing flanker processing but also some evidence for the orienting account: (1) In the DDM, for post-error trials drift rate is reduced suggesting a decreased signal-to-noise ratio of incoming evidence, which is compatible with the orienting account. However, at the same time the weighting of flanker evidence is even more reduced, i.e., flanker influences are suppressed as a net effect. The latter finding is compatible with the adaptive account. (2) On post-error trials, beta power reduction at response latency was diminished more over the ipsilateral compared to the contralateral motor cortex. This suggests that less evidence for the flanker-driven incorrect response is accumulated. (3) Whereas a post-error orienting reflex would predict general attentional disengagement which should be reflected in less steep BPL slopes even on congruent trials (which are unconfounded by distracting evidence), BPL time courses on congruent trials revealed no difference in evidence accumulation in post-error vs. post-correct trials at the time of flanker presentation. Later towards the response, post-error BPL slope is even increased. Given the large statistical power of our study, this non-difference in early evidence accumulation speed favours adaptive error processing accounts and speaks against a significant effect of an error-evoked orienting reflex on attentional control of simple stimulus processing.

In sum, we found more evidence for adaptive accounts of post-error adjustments, but notably we cannot rule out that the post-error response threshold increase actually results from an orienting reflex. While the reduced drift-rate may suggest a general disengagement from attention compatible with the orienting account, the selective suppression of flanker evidence can only be explained by adaptive accounts. Thus, our results show a more detailed picture of several, in part, mutually counteracting effects than a number of previous studies reporting deterioration of perceptual processing after errors[10,19,20,47]. The observed net effect of flanker suppression and PIA is in line with a recent study showing post-error suppression of distracting perceptual information[48]. We found evidence for two mechanisms of post-error adjustments: an increase in decision thresholds, which slows decisions and increases accuracy, as well as an additional shift of evidence accumulation with reduced input from distractors. As a net effect, participants show PES, PIA, and PERI. Given previous work it appears conceivable that the recruitment of adaptive mechanisms and the deteriorating

contributions of the orienting response depend on task demands and context. Interestingly, deterioration of perceptual processing after errors was observed in perceptual decision-making tasks with high uncertainty, in which top-down attentional adjustment enhancing processing in perceptual cortices quickly reaches a ceiling beyond which only a prolongation of evidence accumulation can improve accuracy. In contrast, in a flanker task, accuracy may be improved by a different mechanism—narrowing the focus of spatial attention to the target[49]. Van der Borght (2016)[47] also used a flanker task, and found attentional deterioration in a secondary visual discrimination task. Yet, post-error attentional adjustments related to flanker errors do not need to generalize to other probe stimuli and tasks. Thus, in sum, we argue that both adaptive and maladaptive mechanisms can be triggered by errors and that their relative contributions to behavioural adjustments depend on context and task demands.

More generally, the striking similarity of the evidence accumulation time-courses of BPL and DDM simulation, which was purely fit to RT distributions and accuracy, emphasize the utility of pre-response and peri-response central beta power recordings. While other EEG measures of decision formation have been proposed, such as the P3[41,50], these measures quantify only the absolute amount of accumulated evidence, but not its direction in favour or against each possible response. We consider BPL a validated neural measure not only of decision outcomes, but their temporal evolution integrating incoming information, that furthermore is obtainable from non-invasive surface measures. BPL is of great value for the study of cognitive control processes and decision-making in general, because adaptive changes may not necessarily occur at the level of input selection in sensory regions[51]. Furthermore, regions deemed critical for decision formation may rather be epiphenomenal[52], thus identification of common integratory signals is highly important.

## Methods

**Participants**. Eight hundred ninety-five healthy human subjects (mean age 24.2 years, range 18–40) had given written informed consent to participation. Exclusion criteria for the study were: any history of psychiatric and/or neurological disease; regular use of medication; relevant (consumption within the last month or more than five times in lifetime) history of drug abuse (without cannabis); regular (more than one per month) consumption of cannabis; alcohol intake at day of study; caffeine consumption less than three hours before experiment. All study procedures were approved by the ethics committees of Radboud University of Nijmegen (ECG04032011), where 388 datasets were collected, and the University of Leipzig (285-09-141209), where all other datasets were collected. All procedures were carried out in accordance with the approved study protocol.

Using the proportion of missed trials and trials with more than one response, we tested for non-compliant participants using Grubbs' test for outlier detection with a one-sided alpha of 0.01 (missings: cut off >16%; found in $n = 9$ subjects; multiple responses, i.e., consistent button pressing: $n = 12$). Following inspection of the EEG data, 11 additional subjects were excluded for technical reasons (see below) and the final sample consisted of 863 subjects (434 female).

**Task**. We employed a speeded arrow-version of the Eriksen flanker task. Participants responded as quickly and accurately as possible according to the direction (left or right) of a centrally presented arrow (target) that appeared briefly (33 ms) on the computer screen (Fig. 1). Eighty-three milliseconds prior to the target, four flanking arrows (flankers) appeared above and below the target inducing a tendency to respond in their direction, therefore increasing the likelihood of error responses. The size of all arrows was 1.9° × 1.3° of visual angle and the complete task consisted of 1088 trials. On 50% of all trials, the direction between flankers and target was identical (congruent trials) whereas they pointed in the opposite direction on the other half of trials (incongruent). The distance between the four flankers and the target was modulated in two conditions. In the far condition, the flanker-target distances were 6.5° and 4° visual angle, and in the close condition 3.5° and 1.75°. The time between response and onset of the next trial (response-stimulus interval, RSI) was also modulated in two conditions (short = 250 ms and long = 700 ms). Congruency and flanker-target distance and their respective transitions were counterbalanced in pseudorandom order. Half of congruent trials were preceded by a short and the other half by a long RSI (same for incongruent trials), and additionally half of far trials were preceded by a short, and the other

half by a long RSI (same for close trials). For additional details of the task structure see ref. [32].

**Behavioural analyses**. We determined critical factors that influence RT and accuracy in the task in two multiple robust regression models (GLMs 1 and 2) onto each participant's single-trial accuracy and log-scaled RT. The RT model included the following regressors:

$$\log(RT) = b_0 + Incongruence \times b_1 + Error \times b_2 + Distance \times b_3 + RSI \times b_4 + \\ Previous\ Accuracy \times b_5 + NLD \times b_6 + Trial \times b_7 + e. \quad (1)$$

The logistic accuracy model is given by:

$$Accuracy = b_0 + Incongruence \times b_1 + Distance \times b_2 + RSI \times b_3 + \\ Previous\ Accuracy \times b_4 + NLD \times b_5 + Trial \times b_6 + e. \quad (2)$$

The individual factors are: $Incongruence$ = incongruence between flanker and target ($-1$ = congruent, $1$ = incongruent), $Error$ = accuracy ($-1$ = correct, $1$ = error), $Distance$ = distance between flanker and target ($-1$ = close, $1$ = far), $RSI$ = response-stimulus interval from the previous response ($-1$ = short = 250 ms, $1$ = long = 700 ms), $Previous\ Accuracy$ = accuracy of immediately preceding trial ($-1$ = correct, $1$ = error), $NLD$ = negative log-distance in trials from last break (reflecting changes between breaks within each block of 200 trials), $Trial$ = log-scaled trial number (reflecting the time on the task). $NLD$ and $Trial$ served mainly to control for unspecific effects of task duration, like fatigue or possible adjustments in speed-accuracy trade-offs over the task or each block. Individual participants' $t$-values per regressor were then tested on group level via two-sided $t$-tests against zero. We show follow-up analyses with 99.9% CI in respective figures (Fig. 1c–g, i, j). In these plots, we excluded contributions of possible confounds, for example post-error trials for the analysis of accuracy effects on RT (Fig. 1c).

**Drift-diffusion model and model comparison**. We used a multi-stage sequential sampling model to simulate participants' correct and incorrect RT distributions and the decision process giving rise to these. Specifically, we chose a DDM because it has been shown to reflect the overall behavioural pattern in the Flanker task before[53]. This discrete DDM simulates decisions as a Wiener process with stepwise increments according to a Gaussian distribution with mean $v$ (called drift rate) and variance $s$ on every trial. $v$ reflects the speed of evidence accumulation and $s$ the system's noise, which is commonly fixed (here to 0.1) and scales all other parameters. The step size for all models in which calculations were performed was 1 ms. A decision (response) is triggered when the diffusion reaches a criterion (boundary, determined by the free parameter $\pm a$). Because we had an equal number of left and right responses, we assumed that there was no bias in response selection over the task. However, individual trials were allowed to start with a bias towards one response (start-point variability, parameter $sz$). We used symmetrical boundaries which were defined as left-hand responses when the positive boundary was reached first, and as right-hand responses when the negative boundary was reached first (see Supplementary Methods and Supplementary Figure 1 for more details). The non-decision time was modelled as another free parameter $T_{er}$.

The stages of the model per trial were defined as a zero-mean baseline until flanker onset, a noisy diffusion with $v = 0$ during the non-decision time, a diffusion driven by the flanker direction between flanker and target onset (83 ms) with drift = $v_t \times f_t$, and the target phase thereafter with drift = $v_t$ and the direction of the target. Single-trial values $v_t$ and $f_t$ were determined according to Gaussian variance parameters $sv$ and $sf$. For display purposes, in Figs. 5 and 6, we modelled a consecutive return of the decision variable to baseline similar to an Ornstein-Uhlenbeck process to facilitate comparisons between model and BPL. To speed up the model fitting procedure, we did neither simulate baseline periods nor return to baseline during the fitting because these have no effect on model predictions.

We compared three different variants of the DDM by fitting their parameters to RT and accuracy data observed in the group of 863 participants using quantile maximum likelihood statistics[39] and differential evolution algorithms[54]. Additionally, we used a mixture model assuming 2% outliers that were distributed uniformly over the full range of RTs in correct and error responses. This down-weights the impact of possible outliers on model parameters.

DDM 1 used the same drift rate during flanker and target processing, thus not allowing for suppression of distractors (parameter $f$ fixed to 1). DDM 2 fit parameter $f$ which suppressed flanker processing when below the value of 1. DDM 3 furthermore included trialwise variance in $f$ modelled as a zero-mean Gaussian distribution with variance $sf$. As we found that DDM 3 provided the best fit to the data according to approximate BIC (see Supplementary Methods), we used this model to separately investigate model parameters when the data was restricted to post-correct and post-error trials (DDMs 3.1 and 3.2, respectively). For these fits, variance parameters were fixed to the values of the group mean maximum-likelihood parameters for DDM 3 (displayed in Fig. 2a). In sum, the full model (DDM 3) comprised 4 free parameters ($v$, $f$, $a$, $T_{er}$) and trialwise variance ($sv$, $sf$, $sz$, $st$) which were partly fixed for analyses of less complex models (DDMs 1 and 2) which are thus nested within DDM 3. See Supplementary Table 1 for parameter values.

Two similar variants of DDMs have been implemented to model behavioural effects in conflict tasks that both assume modulations of visual selectivity in multiple stages[53,55]. In accordance with these models, we found that multiple stages (reflecting differences in selective attention) are required to describe the behavioural data adequately. The main reason for this is that a single-stage DDM can not easily fit conditions like incongruent trials which induce slow correct but fast incorrect responses. However, in order to keep the model as simple as possible, we assumed that the stages are defined by the time stimuli are presented on screen rather than two separate processes reflecting stimulus and response selection[55] or a spotlight that changes over time and reduces the influence of flanker stimuli[53]. Thus, our model is very similar to these previous DDM versions and provides an easy means to comparing modelled and neural signal time-courses as well as parameter values representing attentional modulations between conditions.

**Across-participants analysis of model parameters**. We tested which model parameter was individually associated with accuracy, speed of choices, the size of a participant's interference effect, and the ratio of errors made in congruent compared to incongruent trials, across the group of participants using the following regression model (GLM 3):

$$Participant\ factor = b_0 + v \times b_1 + sv \times b_2 + a \times b_3 + sz \times b_4 + \\ Ter \times b_5 + st \times b_6 + f \times b_7 + sf \times b_8 + e. \quad (3)$$

Results for each parameter are plotted as regression weights with their associated SE and displayed in Fig. 2g–m.

**Within-model regression of parameters for speed and accuracy**. This analysis was performed to investigate the relative contribution of trial-wise variance parameters to accuracy and speed of modelled responses according to the following formula:

$$Model\ accuracy\ /\ Model\ RT = b_0 + SP \times b_1 + SP\ Resp \times b_2 + SP\ Abs \times b_3 + \\ DriftRate \times b_4 + Flanker\ weighting \times b_5 + Ter \times b_6 + Incongruence \times b_7 + e. \quad (4)$$

$SP$ reflected the start-point of the diffusion process on every given trial. $SP\ Resp$ is the same value but rectified towards the decision of the model. I.e., if the model chose the positive response on a given trial, a positive value of $SP\ Resp$ per trial would indicate that the diffusion started closer to the chosen boundary. $SP\ Abs$ reflected the absolute bias towards any response per trial. The other factors reflect the variance in $v$ (depending on $sv$), $f$ (depending on $sf$) and the incongruence of each trial. The analysis was based on simulation of 5.000 trials using group mean maximum-likelihood estimated parameters from DDM 3.

**Parameter comparison between models**. To compare individual contributions of parameter changes for models DDM 3.1 and 3.2 (fit to post-correct and post-error trials, respectively), we used the following logistic regression:

$$TrialType = b_0 + v \times b_1 + a \times b_2 + Ter \times b_3 + f \times b_4 + e. \quad (5)$$

$v$, $a$, $T_{er}$, and $f$ reflected the free parameters that were individually fit.

**Simulation of the modelled decision variable**. We used the mean maximum likelihood parameters from the group fit obtained for DDM 3 to simulate the temporal evolution of the modelled decision variable for Fig. 5. For Fig. 6, we used DDMs 3.1 and 3.2 to compare post-correct and post-error traces of the evolution of the decision variable. For all simulations, we computed 5000 simulated trials.

**EEG processing**. Electroencephalographic signals were continuously recorded at 500 Hz (BrainAmps MR plus, BrainProducts) from 60 Ag/AgCl sintered electrodes arranged in the extended 10–20 system in elastic EEG caps (EasyCap). Impedances were kept below 5 kΩ. Analyses were performed using EEGLAB 13.5[56] and custom code written in MATLAB 2015a and 2016a (MathWorks). Ocular channels were positioned at the left and right outer canthi and above and below the left eye. An electrode placed over the sternum served as the ground electrode. The signal was online referenced to A1 and offline re-referenced to common average. Data were offline band-pass filtered between 0.5 and 42 Hz and epoched from $-1.5$ to 2 s relative to target onset. Epochs contaminated with artefacts were automatically rejected based on signal outliers with a dynamically adjusted rejection threshold to remove at least 1 trial separately for error and correct responses and maximally 5% per condition (average number of rejected epochs: 41, range 11–53). Epochs were then demeaned and submitted to adaptive mixture independent component analysis (AMICA)[57]. Independent components that reflected stereotypical artefactual signals such as eye blinks were identified using a correlation-based approach[58]. Additionally, three researchers acquainted with EEG analyses (AGF, CD, TK) inspected components automatically identified and added other, less homogenous artefact components to be removed from the signal (average number of removed components = 3.4, range 1–14). EEG datasets for which ICA did not converge or

with broken central channels were excluded (11 subjects). Stimulus-locked epochs were re-aligned to response onset spanning from −1 to 1 s.

**Response-related power changes**. To confirm that sensorimotor beta power reflected response-related signals and that this was specifically lateralised to the active motor cortex, we convolved the artefact free stimulus-locked EEG signal, baseline corrected from 1.1 to 0.9 s before stimulus onset at all electrodes, with a series of complex Morlet wavelets between 4 and 25 Hz. We used 20 linear steps and a wavelet width of six cycles. Data were then log-transformed before re-epoching to response onset. We firstly confirmed that beginning around 300 ms prior to the response, an overall decrease in beta power relative to a baseline (−900 to −600 ms prior to response) was present, which was followed by a consecutive increase in beta power (beta rebound) around 400 ms post response (Fig. 5a). To analyse whether this signal was specifically lateralised to the hemisphere initiating a response (i.e., contralateral to the responding hand), we used single-trial multiple robust regression with the following model:

$$Beta\ power = b_0 + Hand \times b_1 + Incongruence \times b_2 + Distance \times b_3 + RSI \\ \times b_4 + Following\ RSI \times b_5 + Trial \times b_6 + \log(RT) \times b_7 + e. \quad (6)$$

Here *Hand* reflected the response hand (−1 = left, 1 = right), log-RT the log-transformed RT of the trial, *Following RSI* the next RSI (−1 = 250 ms, 1 = 700 ms), and *Trial* the current log-transformed trial number to account for possible changes over the task, which may be pronounced during early stages of the task. The model was regressed onto the signal at every sample point, electrode and frequency per participant to derive beta weights for factor *Hand* while accounting for unspecific task effects. This analysis confirmed that, beginning around 100 ms before response onset, beta power decreased over the sensorimotor cortex contralateral to the response, i.e., there was a positive covariation between Hand and the signal in the beta band seen at C4/CP4, and a negative covariation at C3/CP3 (Fig. 5b). The signal decreased more at C3/CP3 compared to C4/CP4 when the response was given with the right hand, and, vice versa, it decreased more at C4/CP4 compared to C3/CP3 when the response was given with the left hand. Thus, the signal decreased whenever a response was given with the contralateral relative to the ipsilateral hand. Additionally, an opposite effect was apparent for lateralisation in the theta frequency band (4–8 Hz) at the same electrodes, which is not further interpreted in this report. The topography plots suggest a source compatible with the primary motor-cortices, but it may extend more posteriorly as was similarly found in a study that used MEG and MRI to aid source reconstruction[27]. Next, we collapsed regression weights in the beta band (13–25 Hz) and analysed the scalp topography around the time of response (Fig. 5b). The choice of frequency bands was empirically motivated by including a symmetrical range of frequencies around the peak effect of effector-specific spectral activity. The effect was most pronounced over postero-central electrodes over the sensorimotor cortices (C3, C4, CP3, CP4). We chose these electrodes for all further analyses of *lateralised* beta power (BPL).

**Calculation of lateralised beta power**. To derive measures of lateralised and mean beta power, we collapsed the convolved signal across the frequency range of 13–25 Hz, down-sampled to 250 Hz, and normalised within each participant by dividing the power by its SD and subtracting the mean. We then subtracted beta power over the inactive sensorimotor cortex (i.e., the electrode side ipsilateral to the hand that gave the response in the trial) from the beta power recorded over the active (contralateral) sensorimotor-cortex. This difference signal thus compares the degree of beta power reduction between both hemispheres, presumably reflecting differential motor activation (Fig. 5b).

Although we did not exclude any participants from the analysis post hoc (to keep the sample as representative as possible), we tested in how many participants beta power reduction was lateralised to the executing motor cortex. We found that beta reduction was larger on the active compared to the inactive hemisphere at the selected electrodes in 790 out of 863 participants. All reported results remained significant when the sample was reduced to those participants demonstrating lateralisation of beta power reduction upon response, i.e., excluding 73 participants (8.5% of the sample).

To test single-trial associations of beta thresholds, we used the response-locked single-trial signal (mean ± 12 ms) and regressed behaviourally relevant factors onto the beta threshold according to the following equation:

$$BPL\ Threshold = b_0 + Incongruence \times b_1 + Error \times b_2 + Distance \times b_3 + \\ RSI \times b_4 + Previous\ Accuracy \times b_5 + NLD \times b_6 + Trial \times b_7 + \\ \log(RT) \times b_8 + e. \quad (7)$$

We used the same model to test the separate contributions of each hemisphere (ipsilateral and contralateral) and to investigate pre-stimulus effects in the time-range between −100 and 0 ms (Supplementary Figure 5).

**Data availability**
All data and computer code are available upon reasonable request from the authors. Requests should be addressed to A.G.F. (adrian.fischer@ovgu.de).

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

## Acknowledgements

This research was supported by the Deutsche Forschungsgemeinschaft (DFG; T.A.K. was supported by KL 2337/2-1; M.U. was supported by the Collaborative Research Center SFB779 Neurobiology of Motivated Behaviour, TP A12) and A.G.F. was supported by a grant from the CBBS ScienceCampus financed by the Leibniz Association (SAS-2015-LIN-LWC). The authors thank M. Dreyer for her help with data acquisition and B. Franke and G. Fernandez for their support in recruiting the Nijmegen sample. Furthermore, we would like to thank Felix Molter and Rasmus Bruckner for discussions on the DDM implementation and proofreading.

## Author contributions

C.D., T.A.K. and M.U. designed the study, T.A.K. and M.U. organized data collection, A.G.F. conducted the analyses supported by R.N. and M.U. A.G.F. and R.N. wrote the main manuscript text. All authors reviewed the manuscript.

## Additional information

**Competing interests:** All authors declare no competing interests.

