## [Peer Review File · Nature Communications]

Reviewers' comments:

Reviewer #1 (Remarks to the Author):

In this paper Fischer et al test two alternative accounts of the long studied phenomenon of post-error slowing: the adaptive error processing account (where stimuli are processed more efficiently and more carefully) versus an orienting response account (in which stimulus processing is transiently disrupted by error detection). To this end the authors administered a flanker task to a very large group of participants with concurrent EEG recordings. Regression models are used to disentangle factors such as stimulus congruence, preceding choice accuracy, flanker distance, response to stimulus onset delay and demonstrated significant levels of post-error slowing alongside post-error increases in accuracy and decreases in flanker interference. EEG analyses focussed on lateralised pre-motor beta-band activity and generally appear to support the adaptive processing account of PES, with beta activity reaching a higher level at response execution following an error and exhibiting diminished susceptibility to flankers all of which accords with the observation of increased post-error accuracy.

General Impression:

This is a timely study that stands to have a significant impact on our understanding of post-error slowing. The paradigm and analytic approach are all well suited to providing a greater mechanistic understanding of this phenomenon than has previously been available despite decades of research. While I feel that there are several important results reported in this manuscript that will draw interest from the community I do have a number of substantial concerns pertaining to the methodology and, most particularly, the interpretation of results. I am confident that these can be addressed in a revision.

Major Comments

The authors center their analyses on lateralised beta activity in order to test for post-error adjustments to decision threshold, evidence accumulation rates and distractor interference. However, the focus on a lateralisation index which reflects the difference in amplitude of the contralateral and ipsilateral means that there are several different ways that cross-condition differences in lateralisation at response could be interpreted. For example, rather than there being any difference in the 'threshold', the reduced lateralisation on incongruent vs congruent trials could simply reflect that incongruent flankers drive increased preparation of the ipsilateral response resulting in a smaller lateralisation index. Similarly, post-error increases in lateralisation could potentially reflect diminished ipsilateral preparation. So to provide a more definitive test of their threshold increase interpretation the authors should also measure amplitude at response for the contralateral beta signal in isolation. This is a critical point because the authors' interpretation of increased motor thresholds would actually be at odds with observations from single-unit and functional neuroimaging studies which generally report that only the starting level of motor activity (i.e. before stimulus onset) varies with experimental demands while the motor threshold is often invariant. Thus the authors should also examine whether the amplitude of the separate contralateral and ipsilateral beta signals at stimulus-onset varies post-error versus post-correct. Note that for certain analyses the lateralisation index does seem ideal. I am thinking here specifically of the analysis showing that motor preparation is initially influenced by the flanker direction before diverting towards the correctly chosen direction.

The topographies of the beta signals are unusually posterior. Can the authors comment on this? The authors focus on the 13-25Hz range, what led them to select this specific range? For example De Lange et al (2013, J Neurosci) used the 8-30Hz range while O'Connell et al (2012, Nat Neurosci) used 16-30Hz.

If I have understood the task design correctly, the epoch for analysing beta activity extends back

to the previous stimulus (-1.5s). To what extent can we be sure that post-error beta traces are not influenced by error-evoked responses?

Lines 247/248 higher RT was associated with 'decreased decision boundaries'. No illustration of this effect is provided. Again, measuring adjustment to the separate contralateral and ipsilateral signals would be beneficial here.

Lines 290-296 I find this section very difficult to understand, I am not sure what the authors are trying to say here. Isn't the longer continuation of evidence accumulation following mistakes simply consistent with the observation of increased RT on those trials? I do not understand why this results suggests that flanker effects were reduced and counteracted by sensory input from the target. If the threshold really is increased on post-error trials then, by definition, we should see evidence accumulation for a longer period of time, without calling for any difference in selective attention.

Beta rebound analysis. I am not convinced by the author's interpretation of the result. A more parsimonious explanation is that the reported effects reflect choice history biases. If participants on average have a bias towards alternating their choices from one trial to the next then one would expect elevated motor preparation for the favoured alternative and reduced preparation for the unfavoured one. This would impact on the rebound in a lateralisation index, with a larger rebound reflecting a stronger bias towards switching the response and leads to the prediction that stronger rebound will result in faster RTs when responses were switched versus when they were repeated. Again, this is something that could be examined by comparing contralateral and ipsilateral responses. However, in my view the beta rebound analysis seems peripheral to the purposes of the study and the Results section is already quite dense and at times difficult to follow due to the number of different factors that must be parsed. Therefore my recommendation would be to remove this analyses and perhaps examine it further for a separate publication.

Line 376,377. The authors note that their findings are at odds with previous studies. I think it is important here to acknowledge that the behavioural and electrophysiological findings of the present study are likely to be paradigm specific to a certain degree. For example, Purcell & Kiani's (2016, Neuron) data appear to give strong support to a combination of diminished sensitivity and threshold increases driving post-error behaviour. This observation is not necessarily at all in conflict with the present findings given the marked differences in paradigms. But this is a critical point that the authors would do well to highlight and discuss.

Lines 398-400. I do not understand the point the authors are making here. How can we be sure that beta activity is not epiphenomenal as well? To the extent that, like LIP neurons, beta band provides a readout on the decision process, it doesn't matter much whether the signal is epiphenomenal or causally involved in triggering choices.

Minor Comments

At the end of the introduction, where the authors state that a flanker task was administered (line 77) it should be mentioned here that the task included manipulations of flanker distance and response to stimulus-onset delay as well as congruence.

At the outset of the Results the authors state that the behavioural data were analysed in GLM1 (line 100). Instead they should state here the nature of GLM1 to give the reader a better sense of how the analysis was conducted without necessitating switching to the Methods. Bascially likes 458 to 460 would do the job.

Line 341. The topography of the beta effect is consistent with an origin in motor cortex. The

topography is centro-parietal so this hardly corresponds to motor cortex. At best premotor cortex is the likely source. The authors should be more specific and might consider previous work examining the intracranial origins of action selective beta band activity.

Line 339 and 340. I do not feel that this claim is really warranted as far more comprehensive analyses of the relationship between beta band dynamics and choice behaviour have been reported in previous studies. I think the authors would do better to lean on this previous work to motivate their use of beta as a signature of decision formation since only a few of the essential tests of this hypothesis are provided in the current paper and, in particular, sensory evidence strength is not manipulated in a readily interpretable way.

Lines 415 to 421. Again I am not clear on what point the authors are making here. I would suggest rewriting this section or removing it.

The authors repeatedly refer to the ipsilateral sensorimotor cortex as 'inactive'. This is not an appropriate term because, although there is greater activity for the contralateral vs ipsilateral cortex at response execution this does not mean that the ipsilateral cortex is silent or inactive and there is ample evidence to the contrary. So I would drop this terminology.

Reviewer #2 (Remarks to the Author):

The authors provide a thorough decomposition of EEG activities during behavioral adjustments on a flankers task in a very large (>800 participants) sample. The authors address a controversy in this field: are post-error adaptations reflective of controlled behavior, or are they simply slowed as a tangential feature of an underlying orienting response?

The methods are well-done and the interpretations benefit from the large sample which absolves any ambiguities between meaningful and non-meaningful effect sizes. However the writing is at times very dense and it relies on the reader applying a meaningful amount of effort and memory to keep all the hypothesis tests, statistical tests, factors, measures, and interpretations of the measures straight. This aspect of measures and interpretations is where I think the manuscript is most problematic.

The authors interpret the EEG data using very confident descriptions of latent computational processes, some of which are not fully supported by the manuscript. My major concern will revolve around sub-issues in this approach:

1) The operationalization of the psychological constructs is tenuously associated with the manifest EEG indicators. This concern is partially due to the structure of the manuscript: some constructs become well-associated with EEG activities (e.g. beta peak at response with decision threshold) whereas others do not become fully tested (e.g. beta slope with evidence accumulation, more on that below). However these EEG features are confidently interpreted as faithful reflections of these latent constructs throughout the entire manuscript - prior to evidence that does (or does not) support them. I think it would benefit the manuscript to provide a data-validated model of the manifest-latent relationships first.

2) Regarding evidence accumulation in the beta signal, the authors interpret this as a clear state of evidence accumulation (p.g 5), yet this is highly problematic. I accept the argument that motor cortex could accumulate evidence. Yet other brain areas may be involved as well (e.g. the authors acknowledge the role of P3 in this process in the discussion). The motor cortex activity may result from the output of a separate accumulator, making it definitively epiphenomenal. This possibility is not adequately addressed, and it strongly affects the confidence by which the authors can assume that latent computational features map onto manifest indicators. The authors seem tangentially

aware of this possibility (“pg. 10: regions deemed critical for decision formation may rather be epiphenomenal”), but do not apply this intellectual rigor to their own assumptions.

3) In line with the concern above, the focus of the report solely on motor activities is too limited for the aims that the authors are trying to address. The role of P3 or other accumulators is one example. Yet on pg. 7 the authors aim to address “more evidence for increased attentional control”, which may be most face-validly addressed via visual cortex activity (such as alpha band suppression which is known to be altered after errors, see Rebecca Compton’s work). The authors even suppose on pg. 9 that visual processing changes following errors (“focused on relevant positions following an error”), yet they follow it up with motor cortex signals.

4) I’m not convinced that these latent constructs can be firmly tested without application of a drift diffusion model (DDM), which somewhat tacitly determines the objective definition of these constructs.

a) Pg. 7: “The orienting account show be associated with general slowing of evidence accumulation”. Is this necessarily a prediction of the orienting account? I think it could certainly predict an increase in non-decision time alone, which would be different than a threshold increase that would allow increased accumulation. This would have different implications for downstream processes, and it could certainly be a feature within the adaptive account. Without a DDM, this possibility cannot be tested.

b) I was not fully convinced that the slope of beta power reflects evidence accumulation, epiphenomenon or not. There simply wasn’t enough information and tests of this hypothesis for me to confidently assume this process was occurring during this aspect of the signal. This is critical, as strong theoretical interpretations are made on pg. 9 of the discussion based on this tenuous operationalization (“The slope of BLP indicates the net speed of evidence accumulation in favor of one response”; “this non difference in evidence accumulation speed favors adaptive error processing accounts ...). DDM would help to bolster this assumption.

c) In general, the effects here are tested as being linear expect in the case of statistical moderation. Pg. 5: “Thus, behavioral slowing induced by errors mediates no further accuracy increase as would be expected on comparatively slow trials that were preceded by correct responses”. I’m not sure that post-error responses don’t contain a meaningful degree of non-linearity compared to post-correct trials. As shown by Kiani & Purcell (2016), accumulation and threshold can interact in complex ways following errors; the addition of an orienting response could interact with adaptive process in a variety of manner (see 4a above). I think including a formal DDM account of these data would be one positive step towards being able to make the conclusions that the authors are trying to make here.

Reviewer #3 (Remarks to the Author):

This EEG study investigates the neural mechanisms of post-error adaptation. The study addresses a timely and relevant question. The applied methods are sound, the paper is based on an exceptionally large dataset, is generally well written and presents several interesting findings. Nevertheless, I feel that the conceptual advance of the paper is limited by several shortcomings.

1) I feel that previous studies that already showed key results of the present study are not sufficiently discussed. This includes in particular the general role of beta-lateralization as a marker

of decision-dynamics and the post-error threshold-increase. In particular, Purcell et al. Neuron, 2016 presented not-only strong neurophysiological evidence for a post-error threshold-increase, but also converging evidence for this increase based on computational modeling of behavior (drift-diffusion-model, DDM). Similarly, Dutilh et al, 2012 (DOI 10.3758/s13414-011-0243-2) already presented strong DDM-based evidence for a post-error threshold-increase. In light of these findings, the reported post-error threshold-increase seems somewhat confirmatory.

2) I feel that the most interesting finding of the present study is the claimed absence of a post-error decrease of target processing, which is interpreted as a post-error focusing of attention. This is in contrast to several previous reports (e.g., Purcell et al. 2016). However, this central conclusion is based merely on a negative finding, i.e. the absence of a slope-effect of preceding errors. Thus, this claim should be substantiated by further evidence.

E.g., I miss a comparison of target-evoked responses (ERFs) following errors or correct trials, which could reveal differences in target processing. Furthermore, I miss a computational model of the behavioral data (DDM), which could provide complementary evidence for the authors' conclusions.

Independent from these additional analyses, the key result (slope fit) needs to be better presented. E.g., the fitted linear models should be shown with the measured data in order to assess the goodness-of-fit and to show how constrained these fits are. In fact, from Fig. 4g, I feel that the post-error data is rather noisy and that the slope could well be shallower for post-error as compared to post-correct trials.

3) The stronger BPL for congruent as compared to incongruent trials (Fig. 3g) is unexpected and casts doubt on the BPL at response-time as a reliable indicator of decision-threshold. This needs to be discussed.

First of all, we thank the reviewers for their insightful comments and suggestions for improvement of our manuscript. In our opinion, the adaptations in response to their comments helped to clarify a number of issues and to considerably improve the article.

As suggested by reviewers #2 and #3, we have added a formal drift diffusion model as a framework for our analyses and this has greatly improved conceptual clarity of the manuscript in our mind. We use this DDM to more precisely investigate post-error adaptations. Additionally, we analyse the temporal evolution in the DDM akin to the beta signal to investigate similarities and differences between DDM predictions and the actual neural signal. With this we address the point that the reviewers were not fully convinced that BPL resembles prominent features of a decision variable. We would like to point out that development and fitting of such a DDM is complex in itself because "off-the-shelf" DDM solutions, such as HDDM, do not capture our task. The reason for this is that standard DDMs treat interference trials as trials on which drift rates are low to capture the increased RT. However, lower drift rates per trial cannot produce fast errors. Yet, fast errors are a hallmark of interference tasks. We display and explain this problem in more detail below. Therefore, we had to develop a multi-stage DDM that allowed for different drift-rates during individual trials. We reasoned that participants are able to suppress distractor information to a certain degree, which we found corroborated by a formal model comparison. The winning model comprised 8 free parameters which we fit to each of the almost 900 participants. However, analytical solutions for multi-stage DDMs do not readily exist, thus we had to use simulations of the task. Due to the size of the dataset and the fitting complexity, this required more than 50 million iterations of the simulation of more than 5000 trials each. We are very pleased that the model firstly matched participants behaviour very well with regard to RT on congruent, incongruent as well as error trials. Furthermore, the model closely matched participants' accuracies even depending on stimulus congruence. We additionally provide an in-depth analysis of which model parameter relates to behavioural factors using multiple regression analysis across the whole set of participants (Fig. R3).

In fitting the DDM to post-error and post-correct trials separately, we now provide a much more precise analysis of behavioural post-error adaptation effects. Importantly, the DDM's distractor suppression parameter captured the increased flanker suppression following errors (PERI), which corroborates our findings of reduced flanker processing seen in the lateralised beta signal. Furthermore, non-decision time following errors was unchanged, which further favours an at least partly adaptive account of post-error adaptations.

Moreover, we present additional evidence that beta power can be used as a marker of a decision variable. The DDM made clear predictions about the evolution of the decision variable which was remarkably similar to the BPL signal. Some of these predictions were unforeseen, like differences in pre-stimulus baselines, yet even these predictions were matched by the BPL signal.

Finally, we additionally investigate more precisely which motorcortex (ipsi- or contralateral) contributes to each effect seen in BPL. We discuss the relationship between threshold changes over the contralateral hemisphere and reductions of beta-power decreases over the ipsilateral hemisphere as causes for each effect seen in BPL. This additionally provides more insights into the nature of beta power and its relationship to RT and choice determining factors.

Overall, we feel that the manuscript has greatly benefitted from the additional analyses. In the following, we will comment on the points and arguments made by each reviewer, outline the changes we have made because of those suggestions, and offer our opinion on several points which we feel needed further clarification from our side. The reviewers' comments will be included in our response letter (in italic print for clarity) combined with the responses to the comments (in straight blue print). Unless otherwise specified, the page and figure numbers refer to the revised version of the manuscript and figures used that have been changed are referred to with a capital R. Figures that are included only in the response letter are marked as "Figure Response 1" etc. We marked changes in the manuscript in blue and citations from the manuscript are included in the response letter in red print.

Adrian Fischer (on behalf of all other authors)

Reviewer #1

General:

This is a timely study that stands to have a significant impact on our understanding of post-error slowing. The paradigm and analytic approach are all well suited to providing a greater mechanistic understanding of this phenomenon than has previously been available despite decades of research. While I feel that there are several important results reported in this manuscript that will draw interest from the community I do have a number of substantial concerns pertaining to the methodology and, most particularly, the interpretation of results. I am confident that these can be addressed in a revision.

We thank the reviewer for the assessment that our study can have a significant impact on the field and is timely. We additionally thank the reviewer for the valid criticism and remarks to which we respond in detail below. We feel that their incorporation, especially a more focused discussion of the possible origins of the lateralised signal and what exactly may constitute an equivalent of a response determining threshold, has strengthened the manuscript considerably. We have additionally added a formal DDM account of the RT and accuracy data and corroborate the characteristics of BPL temporal evolution by direct comparison of the modelled decision variable and BPL. This revealed surprisingly detailed similarities which in our view strengthens the hypothesis that BPL is a valid marker of the temporal dynamics of decision processes and therefore can be used to study cognitive control and error adaptation effects. We respond to the remarks below and have reproduced relevant parts of the paper for the reviewer's convenience.

Major

1) The authors center their analyses on lateralised beta activity in order to test for post-error adjustments to decision threshold, evidence accumulation rates and distractor interference. However, the focus on a lateralisation index which reflects the difference in amplitude of the contralateral and ipsilateral means that there are several different ways that cross-condition differences in lateralisation at response could be interpreted. For example, rather than there being any difference in the 'threshold', the reduced lateralisation on incongruent vs congruent trials could simply reflect that incongruent flankers drive increased preparation of the ipsilateral response resulting in a smaller lateralisation index. Similarly, post-error increases in lateralisation could potentially reflect diminished ipsilateral preparation. So to provide a more definitive test of their threshold increase interpretation the authors should also measure amplitude at response for the contralateral beta signal in isolation. This is a critical point because the authors' interpretation of increased motor thresholds would actually be at odds with observations from single-unit and functional neuroimaging studies which generally report that only the starting level of motor activity (i.e. before stimulus onset) varies with experimental demands while the motor threshold is often invariant. Thus the authors should also examine whether the amplitude of the separate contralateral and ipsilateral beta signals at stimulus-onset varies post-error versus post-correct. Note that for certain analyses the lateralisation index does seem ideal. I am thinking here specifically of the analysis showing that motor preparation is initially influenced by the flanker direction before diverting towards the correctly chosen direction.

We do agree with the reviewer that it is of high importance to additionally determine the source of lateralisation differences, which may be attributed to increased beta power over the ipsilateral motor-cortex, decreases over the contralateral motor-cortex, both, or alternatively main effects over each hemisphere may not display the same result as the difference between both hemispheres. We would like to point out that it cannot a priori be assumed that only beta power over the contralateral hemisphere reflects the response threshold, given the well-known contralateral inhibition present in the motor system. Therefore, we consider both hypotheses in the revised manuscript, i.e., that thresholds can be apparent in the difference signal itself, and that thresholds are implemented at the level of the contralateral (executing) motor cortex.

Thus, as suggested by the reviewer, we now further disentangle effects we report for BPL by conducting follow up analyses using ipsilateral BP, contralateral BP and BPL. Furthermore, again in accordance with the reviewer, we agree that such dissociation is relevant only for some of the analyses, especially the response threshold modulation and the baseline effect before stimulus onset, for which we now include a separate analysis in Supplementary Fig. S5. However, we would like to point out that the BPL increase at response execution we report and which is in accordance with a threshold increase, is not present during pre-stimulus periods (please compare pre-stimulus and early post-stimulus time windows in Fig. R7a,c). The difference in threshold develops towards the response only. Therefore, changes in overall beta power, or baseline beta power, cause a global slowing following errors, yet, the (relative) BPL contributes additional information.

In short, the analysis of pre-stimulus baseline power revealed that BPL and ipsi- as well as contralateral BP carry unique information. Crucially, when we assume that BPL reflects characteristics of the DDM decision variable, definable predictions can be tested. In the DDM (Fig. R4 and Fig. R6g,i, reproduced below) we found that start-point variance modulated speed and accuracy. In the analysis of BP before stimulus onset, this was reflected only in BPL, not in ipsi- or contralateral BP. For the analysis of BPL at response execution, we indeed find differential contributions of both hemispheres: the difference seen on incongruent trials is purely due to an increase in activity over the ipsilateral hemisphere, and post-error effects are driven by suppression of the ipsilateral hemisphere. We now acknowledge that additional factors contribute to BPL and discuss these phenomena in the revised discussion. Please see below the reproduction of relevant passages of the revised manuscript:

Figure S5. Single-trial regression on pre-stimulus beta power.

Because we found that the DDM decision variable predicted plausible influences of the state of the decision in the baseline period and during the random noise that was not driven by any stimulus input, we looked more specifically for ipsilateral, contralateral and BPL associations before stimulus onset (-100 until 0 ms around flanker onset). We used the same regression as in Fig. 9 extended by the current trials (future) accuracy. For post-error effects, we found that BPL was unchanged ($p = 1$ corrected), whereas beta power over ipsi- ($t_{862} = 15.9$, $p < 10^{-48}$) and contralateral hemispheres ($t_{862} = 15.1$, $p < 10^{-44}$) was strongly increased following errors which demonstrates that global, yet not BPL unspecifically increases following errors. Interestingly, while ipsilateral BP ($t_{862} = 2.1$, $p = 0.25$) was not significantly related to RT, contralateral BP ($t_{862} = 11.9$, $p < 10^{-28}$) showed positive associations with RT (stronger BP was followed by higher RT), BPL was most strongly associated with the RT of the upcoming choice ($t_{862} = 19.2$, $p < 10^{-66}$). Moreover, while neither ipsi- ($p = 1$) nor contralateral ($p = .15$) BP were predictive of a trials' accuracy, BPL was ($t_{862} = -6$, $p < 10^{-23}$). Note that the significant effect of *Incongruence* does not indicate a spurious result but is a result of mapping data to the given response. It indicates that even before stimulus onset the association between the later selected response and BPL is stronger in incongruent compared to congruent trials ($t_{862} = -15.6$, $p < 10^{-47}$). The same is seen in the DDM, where start-point variability influences decisions more strongly when the trial is incongruent as the decision boundary is reached early during the time of flanker processing. These results demonstrate that BPL is a measure that cannot be reduced to the simple analysis of both hemispheres alone because neither was associated with upcoming accuracy.

All follow-up plots show raw BP / BPL data (insets on the right), error-bars reflect 99% CI for regression weights, and 99.99% CI for raw data. Results are adjusted for multiple comparisons via Bonferroni correction. NLD reflects the log-distance from the last break in the task, and trial number the general time on task.

Main manuscript:

Furthermore, when we analysed BPL as well as the separate contributions of contra- and ipsilateral hemisphere beta-power onto upcoming speed and accuracy, we found that only the relative degree of BP favouring one or the other response was predictive of both choice accuracy and speed often before stimulus onset. This resembles a remarkable similarity between BPL and *start-point variance* that is not seen in the beta signal of any one hemisphere alone (Supplementary Fig. 5). Overall, these striking similarities between modelled signal and BPL suggest that BPL can be used as a marker of decision formation to investigate more complex effects of cognitive control related to error processing.

Figure 9. Single-trial regression on beta at response.

a) Results of a single-trial regression analysis on BPL at response (mean \pm 12 ms surrounding button press), comparing relevant task factors. This analysis was performed to rule out that the results demonstrated for RT and post-error effects result from averaging over trials. (b-d) reflect raw values splits for *Incongruence* (congruent vs incongruent), *RSI* (short vs fast) and *Previous Accuracy* (correct vs error). (f-h) display results of separate regression analyses split by ipsilateral (hemisphere that does not induce the choice on the current trial) and contralateral hemispheres.

Following errors, BPL is larger (more negative) which is caused by a stronger increase in BP over the ipsilateral hemisphere, suggesting that following errors, the unchosen response is more effectively suppressed. Note that in general (Fig. 6a), beta power strongly decreases before responses are executed. When responses are slow, BPL is decreased. This is partly due to less decreases over the contralateral hemisphere, but the effect is much stronger when the relative signal per trial is investigated than when either hemisphere on its own is analysed (h). Furthermore, incongruent trials cause reduced BPL. This latter result is induced by changes over the ipsilateral hemisphere (f) where BP more strongly reduced on incongruent trials, which could reflect the preactivation of the competing response tendency.

Error-bars reflect 99.9% CI, a) displays mean within participants *t*-values, and statistics are results of *t*-tests of individual within-subject regressions against zero. RT in (e) is trichotomised. See Supplementary Figure 5 for a similar analysis on pre-stimulus BP. NLD reflects the distance from the last break in the task, and trial number the general time on task.

Figure 4. How model parameters influence accuracy and RT.

Displayed are regression coefficients of a single-trial regression analysis within the DDM using trial-wise variance parameters. The goal of this analysis was to delineate which parameters have strong impact on whether a trial will be correct (a) or fast (b). Akin to the participant data (Fig. 2), we used logistic and multivariate regression to investigate accuracy and RT, respectively. In both models, a trial's incongruence had a strong impact (yellow, compare to Fig. 2a,e). Variance in the start-point of the decision process of each trial did not overall influence accuracy and RT. However, how much the start-point was biased towards the response the model later on selected, significantly decreased accuracy and RT (factor *SP Resp*). This effect was less pronounced for the absolute bias per trial (factor *SP Abs*). Trialwise faster drift rates significantly decreased RT and much less so accuracy. When flankers were more strongly suppressed (f is lower), accuracy increased and RT slightly decreased. Non-decision time (T_{er}) did not influence accuracy, but longer T_{er} on each single trial led to increased RT. Error bars reflect 99.9% CI, plotted are regression weights for one DDM that employed the group mean parameters for all post-correct trials and simulation of 5.000 single trials.

Figure 6. Stimulus related beta power and simulated decision variable.

(a) The BPL is composed of the activity difference recorded over contralateral (green) and ipsilateral motor cortices (blue). A statistically significant lateralisation towards the contralateral motor cortex is seen around 200 ms prior to response execution (b). The inset shows a regression weight map of the scalp topography of beta power when regressed against the side of motor execution according to GLM 3 (see Methods and Supplementary Fig. 4 for details).

(c,e) Comparison of distractor effects on BPL (c) and modelled decision signal (e) in the DDM during stimulus processing. Distractor effects were seen at very similar times on incongruent (red) compared to congruent trials (blue): distractors pushed BPL and the decision variable away from the correct response. (d) Across participants, the degree of distractor representation in BPL (mean between 340 – 420 ms) positively correlated ($r = .20, p < 10^{-16}$) with the interference effect on RT.

Notably, many features of the time-course and dynamics of BPL were closely predicted by the diffusion variable of the DDM. Non-decision time (T_{er}) reflects stimulus processing before it influences the decision in the DDM and the onset of BPL matches T_{er} plus its variance (dots surrounding vertical dashed line in c and e). The similarities extend to pre-stimulus effects. Here, both signals are lateralised towards the chosen response which is caused by single-trial start-point variance in the DDM affecting accuracy and RT. Strong pre-stimulus bias leads to errors (g) but also quick (based on median splitting) responses (i) in the model. The same pattern is evident in the BPL plots (c,f,h), suggesting that BPL influences speed and accuracy of upcoming responses. In the DDM, this is caused by the causal relationship between start-point variance and accuracy and RT. The striking similarity between diffusion signal and BPL hints at similar relationships for the neural data. Note that the pre-stimulus bias is absent when all trials are plotted without mapping to the executed response.

Shades represent 99.9% CI, the vertical blue and orange lines in c-i reflect the mean RT per condition in humans (top) and model (bottom). Grey background in c,f,h indicate significant time-points between conditions after Bonferroni correction. For details on how the simulation the DDM was performed, see Methods, Supplemental Material and Supplemental Figure 1.

2) *The topographies of the beta signals are unusually posterior. Can the authors comment on this? The authors focus on the 13-25Hz range, what led them to select this specific range? For example De Lange et al (2013, J Neurosci) used the 8-30Hz range while O’Connell et al (2012, Nat Neurosci) used 16-30Hz.*

Regarding the topography of the beta signal we would like to point out that the signal is maximal over electrodes C3 and C4 which can be seen as typical locations regarding motor activation (including for example the lateralized readiness potential). Possibly, the reviewer was surprised by the topography plots which we displayed in Fig. 3 of the original submission and which included plots for beta rebound which is found more posteriorly. As we have removed any discussion of this rebound from the manuscript, as suggested by the reviewer (see below), we now display only the topographies for beta power leading up to the response (reproduced below and now displayed as Fig. R6 and Supplementary Fig. S4). This peak location roughly matches the primary motor cortex close to the central sulcus as a likely generator for our observed beta signal. A recent MEG study of lateralised BP, in which individual participants' anatomical MRI scans were used to aid source reconstruction (Pape & Siegel, Nat Comms, 2016), found that response-related BP decreases extended post-centrally. Thus, it may be that not only primary motor-cortex beta, but additionally more posteriorly located areas contribute to the effect.

We have added a sentence to the manuscript at p. 15 of the revised manuscript that describes this result explicitly:

The topography plots suggest a source compatible with the primary motor-cortices, but it may extend more posteriorly as was similarly found in a study that used MEG and MRI to aid source reconstruction (Pape & Siegel, 2016).

Figure Response 1: Reproduction of Figure R4 illustrating the peak of beta power at response execution.

With regard to the chosen frequency range, we agree that different conventions for beta power changes are used depending on the task at hand. While the study by De Lange et al. (2013) included motoric μ -rhythm (alpha frequency) into their

calculation, O'Connell et al. (2012) refer to two studies that actually use different beta band boundaries than the eventually selected. Both studies use perceptual decision-making paradigms.

Our selection was based on the following rationale: First of all, we abstained from taking lower bands (such as alpha/ μ 8-12 Hz) into the calculation, resulting in a lower border of 13 Hz. Second, it has been shown that motoric beta power shows strong individual differences (Pfurtscheller & Lopes da Silva, 1999). Visual inspection of the lateralized signal suggested a peak frequency around 20 Hz for the simple contrast of response lateralisation without inclusion of any analyses that were aimed at research questions central to the manuscript. In order to capture this peak accurately, we selected the 13-25 Hz range. We do not expect the results to change dramatically if we use the "traditional" range from 13-30 Hz. However, since this would entail a complete re-analysis of all data we consider the data driven approach reasonable. To clarify this approach, we added a short sentence in the methods section explaining our reasoning behind this frequency choice.

From p. 15 of the revised manuscript:

The choice of frequency bands was empirically motivated by including a symmetrical range of frequencies around the peak effect of effector-specific spectral activity.

3) *If I have understood the task design correctly, the epoch for analysing beta activity extends back to the previous stimulus (-1.5s). To what extent can we be sure that post-error beta traces are not influenced by error-evoked responses?*

If this comment refers to the idea that the observed beta effects could be some sort of residual or artefact carried over by the error-signal itself, we would like to point out that error-related EEG components are tightly related to theta (4-7 Hz) and not to beta power changes. Although one study related a combined theta-beta component to post-error slowing (Marco-Pallares et al., 2008), this study quantified beta across both response hands resulting in a central topography, which unlikely captures the same processes we are investigating.

On a functional level, we certainly assume that the observed perturbations are caused by the error but reflect more than a mere spectral echo of the error signal. For the effects we see in BPL reduction at response execution, a relation to error-evoked responses is theoretically possible, however, this would have to be assessed in different ways with respect to possible confounds or overlap of activity which would overload the current paper.

As pointed out in response to point 1 above, there were no changes in BPL during stimulus processing (and even before that) in post-error trials (please see Fig. R7a,c and the additional analyses of pre-stimulus BP reproduced above). Additionally, the early slope of BPL, reflecting the speed of change in BP over time, was also unchanged following errors both for congruent and incongruent trials (Fig. R8c,d, reproduced in response to point 5) below), which further suggests that overlapping error-related activity cannot be the source of the observed effects. Therefore, BPL changes that manifest during distractor processing, compatible with PERI, and differences at response, compatible with threshold increases, cannot be explained by a general error signal overlapping into the next trial. Please additionally compare this to overall beta power (ipsi- and contralateral), which is relatively increased following errors, which could be induced by previous trials.

However, a shift in BPL at response execution following an erroneous trial could in general possibly be explained by carry-over effects from the error trial. We further confirmed that the lateralisation was present on trials with long RSI between error and consecutive correct response. Therefore, we confined the single-trial regression model to long RSI trials and repeated the analysis. We found that the effect of factor *Previous Accuracy* ($t_{860} = -7.3$, $p < 10^{-12}$) remained significant, although it was numerically reduced (interaction *RSI x Previous Accuracy* in the full model). This reduction, however, is in accordance with the behaviour: on long-RSI trials there was less PES (see below). We have included this control analysis into the Supplemental Material and reproduce the section below:

From the Supplemental Material:

We additionally tested if post-error threshold increases depended on the time between previous erroneous response and onset of the next stimulus. The purpose of this analysis was to test if these post-error effects were time-dependent in the EEG, given their known time dependence on a behavioural level (Danielmeier & Ullsperger, 2011), and to exclude that post-error effects seen in BPL can be reduced to overlapping activation caused by previous error trials. This is unlikely, because we found no differences in the baseline and early stimulus processing periods in BPL when we compared post-error to post-correct trials (Fig. 7a,c; Fig. 8c,d; Supplementary Fig. 5). If this was the case, we would expect to find it on short RSI trials alone, but not on long RSI trials. A reduction of PERI and effects related to PES is to some degree expected, because both effects decrease over time.

*To this end, we firstly introduced the interaction between *Previous Accuracy x RSI* into the single-trial regression model (GLM 4). We found a weak, but significant, effect here ($t_{856} = 3.9$, $p = 0.001$) indicating that BPL related response threshold increases following errors were stronger on short RSI trials. However, when we confined the model to long RSI trials alone, *Previous Accuracy* remained clearly significant ($t_{860} = -7.3$, $p < 10^{-12}$). Therefore, this data speaks against overlapping error-related effects as the only source of the BPL increase at response level. Rather, this data appears compatible with the notion that PES depends on the time between the error and the consecutive trial. This is further confirmed by an additional behavioural test, in which we compare PES between short and long RSI trials. Indeed, PES is lower on long RSI trials (ΔRT short RSI = 52 ms, long RSI = 20 ms, $t_{862} = -19.7$, $p < 10^{-69}$), compatible with lower BPL threshold increases.*

4) *Lines 247/248 higher RT was associated with 'decreased decision boundaries'. No illustration of this effect is provided. Again, measuring adjustment to the separate contralateral and ipsilateral signals would be beneficial here.*

As in all parts of the manuscript, we have replaced conceptual wordings (e.g., decision boundaries) with a more precise phenomenological terminology. Thus, we do refer only to BPL related to response execution here. As suggested by the reviewer, we have included an illustration of this effect which is displayed in Supplementary Fig. 7e of the revised manuscript (reproduced below). The follow-up analysis for separate contra- and ipsilateral signals is included in Fig. R9f. As we have done for the other parts of threshold analyses, we also added a separate single-trial regression analysis for each of the contributing motor cortices. The plot from Supplementary Fig. 7 is reproduced below and the single-trial regression is reported in response to point 1) above.

Figure S7. Response-locked BPL and DDM decision variable.

Akin to Figure 6, that displays stimulus processing for trials separated by stimulus congruence, accuracy and response speed, we display the same trials here when the analysis is locked to the response. In the BPL signal when comparing correct congruent and incongruent trials (a), we found a significant difference at response execution. Our follow-up analysis of the lateralised signal showed that this effect is due to reduced activation (increased beta power) of the ipsilateral hemisphere on congruent trials compared to incongruent ones. This effect is not modelled in the DDM (b) because it does not include threshold differences between trials.

When comparing correct and error responses, we note that both are associated with similar thresholds in BPL (c) which is comparable to the modelled DDM signal (d). At virtually all other times in the trial, BPL differs between correct and error responses.

As in the stimulus-locked analyses, we find that in BPL (e) as well as modelled signal (f) a lateralisation towards the given response is pronounced for fast responses. Notably, BPL at response execution is reduced for later responses. If BPL is seen as a reflection of response thresholds, this observation would be compatible with dynamically changing decision bounds within a trial (Hawkins, Forstmann, Wagenmakers, Ratcliff, & Brown, 2015).

Note that our diffusion model did not simulate beta rebound (the lateralisation of beta power following response execution). Thus, the model immediately returns to baseline after a decision is made. Plot conventions as in Figure 6.

5) Lines 290-296 I find this section very difficult to understand, I am not sure what the authors are trying to say here. Isn't the longer continuation of evidence accumulation following mistakes simply consistent with the observation of increased RT on those trials? I do not understand why this results suggests that flanker effects were reduced and counteracted by sensory input from the target. If the threshold really is increased on post-error trials then, by definition, we should see evidence accumulation for a longer period of time, without calling for any difference in selective attention.

We apologize for not having explained this complex result with sufficient clarity. In the revision, we have restructured the complete section (reproduced below). In short, we agree with the reviewer that higher RT could be explained by increased boundaries. Yet, both in the DDM as in the BPL signal, we find evidence for reduced distractor processing following errors (consistent with the behavioural PERI effect). This is displayed in Fig. R7c. Because we additionally do not see differences in slope changes of BPL on congruent post-error trials (Fig. R8d), this speaks for a selective suppression of distractors. We reproduce Figure R8 below, that now in more detail explains the interpretation of slope changes:

Figure 8. Beta slope dissociates specific from general adaptations.

BPL slope indexes the steepness of the BPL signal analysed via single-trial fitting of individual periods in the BPL signal surrounding every data point (80 ms). The slope can be seen as the rate of change in BPL, thus reflecting how quickly beta power lateralises to one hemisphere. (a) displays how BPL slopes differ between congruent (blue) and incongruent (orange) trials. The dashed vertical line reflects when BPL slopes start to first significantly deviate from zero, which coincides with the non-decision time parameter in the DDM.

(b) displays that the degree of interference induced by visual proximity of the distractors affects the duration of BPL slopes that point into the incorrect (distractor-) direction. As a result, when incongruent flankers are close to the target, BPL to the wrong direction is stronger and longer-lasting as compared to far flanker trials (cf. Fig. S6). Similarly, for incongruent trials the slope of distractor processing is influenced by previous accuracy (c), which is consistent with PERI. However, there is no significant difference (even at a very lenient threshold of $p < 0.05$) at any point during early BPL slopes when comparing congruent post-error (orange) to post-correct (blue) trials. This suggests that post-error trials specifically reduce the interference induced by distractors, which is seen in the BPL slope analysis at around 290 to 360 ms.

For an additional analysis comparing how slow and fast responses are associated with BPL slope, see Supplementary Fig. 7. Statistical comparisons between conditions are calculated via repeated t -tests corrected for multiple comparisons (a,b) and uncorrected (c,d) and significant time periods are shaded in grey, shades reflect SE.

6) Beta rebound analysis. I am not convinced by the author's interpretation of the result. A more parsimonious explanation is that the reported effects reflect choice history biases. If participants on average have a bias towards alternating their choices from one trial to the next then one would expect elevated motor preparation for the favoured alternative and reduced preparation for the unfavoured one. This would impact on the rebound in a lateralisation index, with a larger rebound reflecting a stronger bias towards switching the response and leads to the prediction that stronger rebound will result in faster RTs when responses were switched versus when they were repeated. Again, this is something that could be examined by comparing contralateral and ipsilateral responses. However, in my view the beta rebound analysis seems peripheral to the purposes of the study and the Results section is already quite dense and at times difficult to follow due to the number of different factors that must be parsed. Therefore my recommendation would be to remove this analyses and perhaps examine it further for a separate publication.

We agree with the reviewer and have removed the analysis of beta rebound from the manuscript.

7) Line 376,377. The authors note that their findings are at odds with previous studies. I think it is important here to acknowledge that the behavioural and electrophysiological findings of the present study are likely to be paradigm specific to a certain degree. For example, Purcell & Kiani's (2016, Neuron) data appear to give strong support to a combination of diminished sensitivity and threshold increases driving post-error behaviour. This observation is not necessarily at all in conflict

with the present findings given the marked differences in paradigms. But this is a critical point that the authors would do well to highlight and discuss.

We thank the reviewer for pointing this out. Indeed, we have rephrased this section to more clearly reflect the fact that our results are mostly compatible for example with the results of Purcell & Kiani. Similar to their study we find in our DDM that after errors the boundary increases and the drift rate decreases. However, in addition to that we find that flanker input is downweighted, which means that distractor evidence is suppressed. This suppression is furthermore supported by the BPL findings. Flanker suppression appears to counteract reduced drift rate, which in turn explains why we found PIA and PERI effects whereas in the Purcell and Kiani study accuracy remains unchanged after errors. Notably, their task did not allow to dissociate both factors easily because the random dot motion task induces uncertainty at the stimulus level but no response conflict as the flanker task. As we discuss in the revised manuscript, performance in perceptual decision making tasks may be at ceiling, i.e., even increasing selective attention to the stimuli may not help to better extract the evidence from the noise. In contrast, in flanker tasks, both distracting and target information are easily accessible at the perceptual level. This enables adaptation at multiple levels of processing, for example through focusing selective attention (i.e. narrowing the “Mexican hat” of spatial attention onto the target position), through weighting the evidence channels (e.g. based on spatial and temporal characteristics, as flankers are presented more peripherally and earlier than targets), or through suppressing/inhibiting the build-up of flanker-induced response tendencies in the motor system. While our study is not designed to disentangle these possible mechanisms by which flanker suppression is implemented, it demonstrates that adaptive control mechanisms are initiated by errors at all.

We have extended this discussion and the new section is reproduced below:

We used the DDM parameters fitted to the participants’ behavioural data and the BPL as neuronal correlate of the response selection process to test the respective predictions of the adaptive and the orienting accounts for post-error adjustments (Table 1). Behaviourally, following mistakes, participants in the task were both slower and more accurate, which is in accordance both with adaptive changes in cognitive control, but might also be explained by the orienting account, if relative performance adjusted for RT is compared. For post-error trials the DDM showed an increased boundary and the BPL yielded increased peak amplitudes, both indicating increased response thresholds. This explains slower and –resulting from longer evidence accumulation– more accurate responses reflected in PES and PIA²³. This finding is in accordance with predictions from both accounts and with previous findings^{23,50}. Furthermore, we found that distractor influence on the response selection was reduced following mistakes. This was demonstrated by a reduced flanker-driven deflection of the BPL in incongruent post-error trials as compared to post-correct trials and was similarly seen in the modelled DDM signal time-courses (Fig. 7). This finding suggests, in line with adaptive error processing accounts, that participants were more efficiently able to suppress distracting information after they made a mistake. This effect constitutes a rapid adaptation that occurs around 380 ms after onset of the next trial. Because trials here were separated by 250 or 700 ms, in less than a second, stimulus processing changes and is focussed towards relevant positions following commission of an error. However, under certain circumstances, an orienting reflex might yield a similar reduction of the incongruence-effect on BPL: Disengagement of attention¹³ could generally reduce perceptual processing of stimuli which should result in slower evidence accumulation. Indeed previous studies found evidence for a reduced signal-to-noise ratio in the perceptual signal²³ and a reduced visual P1²² after errors in perceptual decision making tasks. Thus, a post-error orienting reflex should be associated with slower evidence accumulation which, in turn, could reduce the distracting effect of flankers on BPL. This seemingly specific effect on flankers would be particularly strong, if the effects of the orienting reflex onsets early and decays quickly as has been suggested previously^{24,39,51}, because flanker presentation preceded targets. In contrast, adaptive accounts suggest post-error increases in selective attention^{19,52} or inhibition of conflicting response tendencies¹⁸ which could be reflected in enhanced drift rates and/or relative suppression of flanker-driven evidence compared to target-driven evidence. In sum, DDM and BPL data provide more evidence for adaptive post-error control up-regulating selective attention and selectively suppressing flanker processing but also some evidence for the orienting account was found: (1) In the DDM, for post-error trials drift rate is reduced suggesting decreased signal-to-noise ratio of incoming evidence, which is compatible with the orienting account. However, at the same time the weighting of flanker evidence is relatively reduced, i.e., flanker influences are suppressed as a net effect. The latter finding is compatible with the adaptive account. (2) On post-error trials, beta power reduction at response latency was diminished more over the ipsilateral compared to the contralateral motor cortex. This suggests that less evidence for the flanker-driven incorrect response is accumulated. (3) Whereas a post-error orienting reflex would predict general attentional disengagement which should be reflected in less steep slopes of BPL even on congruent trials (which are unconfounded by distracting evidence), BPL time courses on congruent trials revealed no difference in evidence accumulation in post-error vs. post-correct trials at the time of flanker presentation. Later towards the response, post-error BPL slope is even increased. Given the large statistical power of our study, this non-difference in early evidence accumulation speed favours adaptive error processing accounts and speaks against a significant effect of an orienting reflex on attention.

8) Lines 398-400. I do not understand the point the authors are making here. How can we be sure that beta activity is not epiphenomenal as well? To the extent that, like LIP neurons, beta band provides a readout on the decision process, it doesn’t matter much whether the signal is epiphenomenal or causally involved in triggering choices.

We agree with the reviewer in that the causal or epiphenominal character of beta power is not the core topic of the paper and have removed this section from the manuscript.

Minor Comments

At the end of the introduction, where the authors state that a flanker task was administered (line 77) it should be mentioned here that the task included manipulations of flanker distance and response to stimulus-onset delay as well as congruence. Thank you, we have added this information to the introduction.

At the outset of the Results the authors state that the behavioural data were analysed in GLM1 (line 100). Instead they should state here the nature of GLM1 to give the reader a better sense of how the analysis was conducted without necessitating switching to the Methods. Basically lines 458 to 460 would do the job.

We added the suggested lines to the beginning of the results section to make it more convenient to understand the nature of the reported analyses.

Line 341. The topography of the beta effect is consistent with an origin in motor cortex. The topography is centro-parietal so this hardly corresponds to motor cortex. At best premotor cortex is the likely source. The authors should be more specific and might consider previous work examining the intracranial origins of action selective beta band activity.

We have changed this sentence to acknowledge that we cannot (using EEG) reasonably well determine the neural source of the observed signal. We note that premotor, motor and somatosensory cortex may contribute to the observed scalp signal. The new sentence in the discussion is reproduced below:

While the spatial resolution of EEG does not provide means to safely assume the origin of the beta signal, effector specific signals have been found in humans in primary and pre-motor cortex extending to the post-central cortex (Donner, Siegel, Fries, & Engel, 2009; Pape & Siegel, 2016) and likewise in intracranial recordings in animals (Stetson & Andersen, 2014; Zhang, Chen, Bressler, & Ding, 2008).

Line 339 and 340. I do not feel that this claim is really warranted as far more comprehensive analyses of the relationship between beta band dynamics and choice behaviour have been reported in previous studies. I think the authors would do better to lean on this previous work to motivate their use of beta as a signature of decision formation since only a few of the essential tests of this hypothesis are provided in the current paper and, in particular, sensory evidence strength is not manipulated in a readily interpretable way.

We have rephrased this paragraph entirely and included many references to previous studies that establish beta power as a clear marker of motor preparation.

Lines 415 to 421. Again I am not clear on what point the authors are making here. I would suggest rewriting this section or removing it.

We have removed this section from the manuscript and agree that it was not central to our argument.

The authors repeatedly refer to the ipsilateral sensorimotor cortex as 'inactive'. This is not an appropriate term because, although there is greater activity for the contralateral vs ipsilateral cortex at response execution this does not mean that the ipsilateral cortex is silent or inactive and there is ample evidence to the contrary. So I would drop this terminology.

Thank you, we have changed the terminology and now refer to contralateral and ipsilateral cortices only.

Reviewer #2 (Remarks to the Author):

The authors provide a thorough decomposition of EEG activities during behavioral adjustments on a flankers task in a very large (>800 participants) sample. The authors address a controversy in this field: are post-error adaptations reflective of controlled behavior, or are they simply slowed as a tangential feature of an underlying orienting response? The methods are well-done and the interpretations benefit from the large sample which absolves any ambiguities between meaningful and non-meaningful effect sizes. However the writing is at times very dense and it relies on the reader applying a meaningful amount of effort and memory to keep all the hypothesis tests, statistical tests, factors, measures, and interpretations of the measures straight. This aspect of measures and interpretations is where I think the manuscript is most problematic.

We thank the reviewer for their suggestions and the constructive criticism. We have restructured many aspects of the paper, removed analyses we felt did not contribute significantly to our central question of error-processing (for example the analysis of beta rebound in hindsight appeared somewhat unrelated to this question), and added complex analyses that however are more easily interpreted and understood. This included the DDM which, as suggested, helps to map the data more clearly onto underlying constructs. Please see our responses below.

The authors interpret the EEG data using very confident descriptions of latent computational processes, some of which are not fully supported by the manuscript. My major concern will revolve around sub-issues in this approach:

1) The operationalization of the psychological constructs is tenuously associated with the manifest EEG indicators. This concern is partially due to the structure of the manuscript: some constructs become well-associated with EEG activities (e.g. beta peak at response with decision threshold) whereas others do not become fully tested (e.g. beta slope with evidence accumulation, more on that below). However these EEG features are confidently interpreted as faithful reflections of these latent constructs throughout the entire manuscript - prior to evidence that does (or does not) support them. I think it would benefit the manuscript to provide a data-validated model of the manifest-latent relationships first.

We thank the reviewer for this suggestion and we have added a DDM including many simulations to the manuscript. We furthermore no longer equalize certain EEG phenomena with latent constructs and carefully describe them phenomenologically in order to avoid preemptive conclusions. We discuss the associations between EEG phenomena and latent constructs now in the Discussion section only. Notably, we additionally simulate the DDM decision variable to inspect the time-course of processing and compare this to the BPL signal. We find striking similarities and features that are non-trivial including start-point variance and flanker suppression. Notably, boundary increases are seen both in BPL and the DDM. We feel that these additional analyses have considerably strengthened the manuscript. Below, we reproduce relevant parts of the paper, please see Supplemental Methods for additional details on how the DDM was coded and implemented.

Figure S1. Illustration of the multi-stage DDM.

Stimuli in the arrow flanker task (a). At t_0 flankers were presented either closer (second screen) or further away (see Fig. 1) from the delayed (stimulus-onset-asynchrony (SOA) = 83 ms) target (third screen). The relative direction between flankers and target determines a trials' congruence. Only the target arrow indicates the correct response (green hand = correct congruent, orange = correct incongruent, red = error incongruent). These task phases are explicitly modelled in the DDM (b) with 5 separate stages. This was done as to model the task as closely as possible.

Stage 1 consists of a pre-stimulus baseline for which the value is determined by the start-point variance modelled as a uniform distribution. We did not assume general bias in start points.

Stage 2 models the non-decision time per trial that varies according to the variance parameter st again assuming a uniform distribution ($Ter_{(t)} = Ter \pm st \times 0.5$).

Stage 3 represents a noisy diffusion in flanker direction for the duration of the SOA. On incongruent trials (orange), the decision process drifts away from the correct decision with drift rate $v_{(t)} \times f_{(t)}$. The trialwise variance parameters sv and sf were modelled as Gaussian distributions with mean v and f respectively. Note that lower values indicate stronger suppression of distractors and $f = 1$ indicates no flanker suppression, whereas values > 1 would indicate stronger flanker compared to target processing.

Stage 4 models the consecutive diffusion back into the correct (left hand) direction with drift rate $v_{(t)}$. Because on incongruent trials the drift in the SOA was opposite to the consecutive drift in stage 4, RTs are longer. Error likelihood increases when the baseline is shifted towards the flanker direction and $f_{(t)}$ is higher on a given trial resulting in an early crossing of decision boundaries (red line). Note that this implies that errors are usually faster on incongruent trials.

Stage 5 models the return to baseline of the decision variable as a half-normal distribution with variance 0.1 and start point 1 that downscaled the drift signal once a decision was made, akin to an Ornstein-Uhlenbeck process. This was done as it is unclear if under speeded conditions a decision process truly terminates when a response occurs or continues to evaluate evidence accumulation for a certain period of time to possibly allow quick adjustments. However, this has no influence on model predictions regarding RT and accuracy and only determines how the diffusion signal behaves *after* a decision has been made which does not influence the start of the next trial.

Grey shades illustrate the distribution of trial-by-trial variance parameters (sz = start point variability, sv = drift rate variability, sf = flanker suppression variability, st = non-decision time variability). $\pm a$ = upper / lower decision boundary, $z_{(t)}$ = start-point per trial, $v_{(t)}$ = drift rate per trial, $Ter_{(t)}$ = non-decision time per trial, $f_{(t)}$ = speed of flanker relative to target diffusion per trial. $R_{err,con,inc}$ = time of error, congruent and incongruent response.

From the main manuscript at p. 4ff:

An extended DDM captures task effects. Simple accumulator models integrate sensory evidence into a decision variable that determines choices and RT. This process has been found to be reflected in single neurons in various cortical areas as revealed by intracranial recordings in monkeys^{42,43}. Many of the recorded neurons reflected the accumulated evidence and triggered a response when a common threshold was reached. A simple variant of these accumulator models is the DDM which assumes that choice options are mutually exclusive. This allows to replace two separate accumulators with one decision variable that reflects the difference between both decision options. On a given trial, the decision can be randomly biased to favour one response (*start-point variance*). Because decisions are triggered when a decision boundary is crossed, the height of the boundary parameter (a) determines how much integration of evidence is required to trigger a response. Finally, visual processing and motor output is integrated in the model by a parameter that reflects the non-decision time (Ter), which in our model can vary from trial to trial (st).

Usually, DDMs assume a constant speed of evidence accumulation within each trial, which is governed by a drift rate parameter (v , Fig. 3a) and trialwise variance (sv). However, in conflict tasks such as the flanker paradigm, evidence can first point into one direction and thereafter reverse. To reflect this, we used a multi-stage DDM. In our model, evidence accumulation followed the flankers' direction during the time they were displayed on screen alone (83 ms). After target onset, evidence accumulation was driven by the direction of the target.

We fit three variants of the model to the distribution of RTs for each participant using QMLE⁴⁴ and differential evolution⁴⁵. The first model was a standard DDM that used the same evidence accumulation rate for distractor and target stimuli. The second DDM included the possibility to relatively downweigh, i.e., suppress distractor evidence (parameter f), and the third model additionally allowed trial-by-trial variance in distractor suppression (sf). Model comparison (Fig. 3b) suggested that the last model provided the best fit to the data indicating that participants suppress distractor information, yet this varies between individual trials. Suppression (rather than amplification) of distractors was confirmed by the value of f , which was significantly lower than 1, where 1 would reflect equal processing of distractor and target information (mean $f = 0.44 \pm 0.03$ (99.9% CI), $t_{862} = 64.3$, $p = 0$ within precision). Overall, the model provided a good fit to the data and matched participants RTs on congruent, incongruent and error trials as well as accuracies in congruent and incongruent trials (Fig. 3c-f).

Association between model parameters and behaviour. We then tested which model parameters were associated most strongly across the group of participants with RT, accuracy, interference and the ratio of errors in congruent relative to incongruent trials (Fig. 3g-m). This revealed that RT was mainly reflected by the drift rate parameter (v) and accuracy by the height of the boundary (a) and strength (f) and variance (sf) of distractor suppression. These latter two parameters additionally covaried with the magnitude of the interference effect on incongruent trials per participant (Fig. 3l) as well as with the ratio of incongruent to congruent errors (Fig. 3m). We use this model to provide a quantitative analysis of post-error effects and make predictions on how such a decision variable should behave when it is analysed for task effects. Notably, the model also predicted errors on congruent trials (Fig. 4c), although these were removed for fitting (see Supplements for details).

Finally, we investigated which variance parameters on a single-trial basis affect the models' accuracy and RT using a similar regression approach as in the behavioural analysis (GLM 3, Methods). We found that both accuracy ($p < 10^{-10}$) and RT ($p < 10^{-33}$) were significantly reduced when a trial was biased towards the response that was later on selected due to variance in start-points (sz). Additionally, the degree of trial-wise suppression of distractor information (determined by factor sf in the model) was strongly associated with accuracy ($p < 10^{-52}$) and less with RT ($p < 10^{-9}$). This analysis identifies that especially variance in start-point and flanker suppression are factors that drive individual trials in the model to be fast or slow and correct or incorrect.

Post-error adaptations in the DDM. Next, we fit the same DDM as before to post-error and post-correct trials separately. We excluded participants with less than 30 valid post-error trials ($n = 15$). Additionally, we fixed the variance parameters of the DDM (sv , st , $StarVar$, sf) to the group mean to facilitate convergence. Note that variance parameters were still in the model, only their value did not vary between participants. The DDM captured RT and accuracy well in both post-error and post-correct trials (Fig. 4a-b, d-e).

Next, we compared differences in parameter values for drift rate, boundary, non-decision time and flanker suppression in post-error and post-correct model fits using multiple logistic regression of parameter values onto which trials' the model was fit to according to eq. 5 (Methods). Positive regression coefficients indicate parameter increases following errors (Fig. 5). We found that drift-rate was decreased ($t_{1691} = -15.3$, $p < 10^{-51}$), boundaries increased ($t_{1691} = 17.5$, $p < 10^{-66}$), and non-decision time unchanged following errors ($t_{1691} = -1.7$, $p = 0.1$). Additionally, flankers were more strongly suppressed ($t_{1691} = -5.3$, $p < 10^{-6}$) in post-error trials. This suggests that slower evidence accumulation began at the same time (no change in Ter) and that distractors were more strongly suppressed following errors. A separate parameter recovery analysis confirmed that our fit method reliably identified model parameters (Supplementary Figure 3).

Figure 3. Drift diffusion model fits and parameters.

Our implementation of the drift diffusion model (DDM) included 8 free parameters (a). Apart from the standard parameters coding for drift rate (v), boundary (a), and non-decision time (Ter), we included variance parameters that determined how much these three parameters varied from trial-to-trial (see Supplementary Fig. 1 for more details about the model). Model comparisons (b), revealed that two additional free parameters for distractor suppression (red) and trialwise variance of distractor suppression (yellow) increased model fit (see Supplementary Fig. S2 for results of worse fitting models). The flanker suppression parameter f modulates flanker processing and scales the drift rate during the time the distractors are on screen. Additionally, Gaussian variance (sf) for this parameter that influenced trial-wise values, further increased model fit (b). (c-e) shows quantile fits of the model (dark blue) against human RT data and (f) shows model and human accuracy. In all conditions (congruent & incongruent correct as well as incongruent error), the model captures the RT data in each quantile, suggesting a good fit to the data. Boxes = interquartile range (IQR), o = median, - = mean, whiskers = $1.5 \times$ IQR, grey dots = outlier. Note that we removed congruent errors from the analysis as these were very rare (<2.5% of trials) and some subjects did not commit congruent errors at all. (g-m) displays the relationship between model parameters and behaviour across subjects. Displayed are regression coefficients and 99.9% confidence intervals.

(g) Faster participants were fit by higher drift rates, lower decision boundaries, and lower non-decision times.

(h-j) RT on error trials (h) was most strongly dependent on the non-decision time because errors are usually very fast, whereas drift rates are more closely associated with RT on correct congruent (i) and incongruent (j) trials.

(k) Accuracy was most strongly reflected in the height of the boundary (a) but additionally dependent on how much distractors were suppressed (f) and how variable this suppression from trial-to-trial was (sf). Higher variance reduces accuracy, because in more trials distractors are likely to not be suppressed.

(l) Interference and the ratio of errors between incongruent and congruent trials (m) additionally covaried with distractor suppression and its variance (f, sf) and the less flankers were suppressed in the model, the higher was the interference and the more incongruent compared to congruent errors a participant made.

Figure 4. How model parameters influence accuracy and RT.

Displayed are regression coefficients of a single-trial regression analysis within the DDM using trial-wise variance parameters. The goal of this analysis was to delineate which parameters have strong impact on whether a trial will be correct (a) or fast (b). Akin to the participant data (Fig. 2), we used logistic and multivariate regression to investigate accuracy and RT, respectively. In both models, a trial's incongruence had a strong impact (yellow, compare to Fig. 2a,e). Variance in the start-point of the decision process of each trial did not overall influence accuracy and RT. However, how much the start-point was biased towards the response the model later on selected, significantly decreased accuracy and RT (factor *SP Resp*). This effect was less pronounced for the absolute bias per trial (factor *SP Abs*). Trialwise faster drift rates significantly decreased RT and much less so accuracy. When flankers were more strongly suppressed (f is lower), accuracy increased and RT slightly decreased. Non-decision time (T_{er}) did not influence accuracy, but longer T_{er} on each single trial led to increased RT. Error bars reflect 99.9% CI, plotted are regression weights for one DDM that employed the group mean parameters for all post-correct trials and simulation of 5.000 single trials.

Figure 5. Model changes following errors.

When we fit the model separately to all trials following correct responses (a,b) and all trials following errors (d,e), we again observed that the model captured RT distributions in all quantiles of the data despite partly limited trials in the post-error condition. By comparing (a) against (d) and (b) against (e) it can also be seen, that the model captured PES. Furthermore, the model displayed significant accuracy increases especially in incongruent trials (c) post-error (t -test $t_{348} = 17.5$, $p < 10^{-56}$).

(f) Model parameter values were compared using logistic regression onto *TrialType* (post-error, post-correct) according to eq. 5. Plotted are regression weights and standard errors for individual factor contributions to the difference between models fitted to post-error and post-correct trials and positive values indicate parameter increases following errors. The largest change in parameter values was observed for decision boundaries, which we found to be increased when the model was fit to post-error trials. Additionally, drift rates were decreased, yet the non-decision time was unchanged. Crucially, the degree to which distractors influenced the decision variable (multiple linear regression $t_{1691} = -5.3$, $p = 1.24 \times 10^{-7}$) was decreased following errors, which was seen over and above decreased drift rates or increased boundaries.

Plot conventions for a-e as in Fig. 3, error-bars in f reflect 99.9% CI.

Figure 6. Stimulus related beta power and simulated decision variable.

(a) The BPL is composed of the activity difference recorded over contralateral (green) and ipsilateral motor cortices (blue). A statistically significant lateralisation towards the contralateral motor cortex is seen around 200 ms prior to response execution (b). The inset shows a regression weight map of the scalp topography of beta power when regressed against the side of motor execution according to GLM 3 (see Methods and Supplementary Fig. 4 for details).

(c,e) Comparison of distractor effects on BPL (c) and modelled decision signal (e) in the DDM during stimulus processing. Distractor effects were seen at very similar times on incongruent (red) compared to congruent trials (blue): distractors pushed BPL and the decision variable away from the correct response. (d) Across participants, the degree of distractor representation in BPL (mean between 340 – 420 ms) positively correlated ($r = .20, p < 10^{-16}$) with the interference effect on RT.

Notably, many features of the time-course and dynamics of BPL were closely predicted by the diffusion variable of the DDM. Non-decision time (Ter) reflects stimulus processing before it influences the decision in the DDM and the onset of BPL matches Ter plus its variance (dots surrounding vertical dashed line in c and e). The similarities extend to pre-stimulus effects. Here, both signals are lateralised towards the chosen response which is caused by single-trial start-point variance in the DDM affecting accuracy and RT. Strong pre-stimulus bias leads to errors (g) but also quick (based on median splitting) responses (i) in the model. The same pattern is evident in the BPL plots (c,f,h), suggesting that BPL influences speed and accuracy of upcoming responses. In the DDM, this is caused by the causal relationship between start-point variance and accuracy and RT. The striking similarity between diffusion signal and BPL hints at similar relationships for the neural data. Note that the pre-stimulus bias is absent when all trials are plotted without mapping to the executed response.

Shades represent 99.9% CI, the vertical blue and orange lines in c-i reflect the mean RT per condition in humans (top) and model (bottom). Grey background in c,f,h indicate significant time-points between conditions after Bonferroni correction. For details on how the simulation the DDM was performed, see Methods, Supplemental Material and Supplemental Figure 1.

Figure 7. Post-error adaptations in beta power and simulated choices.

Plotted are BPL time-courses following error and correct responses for all trials (a,b) and incongruent trials specifically (e,d). In the lower row the simulated decision signal derived from the DDMs fit to post-correct and post-error trials is plotted split up by the same factors (e-h). We found evidence for increased BPL at response execution (b), which in the DDM is caused by boundary increases following errors (f). Additionally, post-error reduction of interference (PERI) is seen in the beta signal (c,d) and predicted by the DDM decision variable as well (g,h). We additionally confirmed that not only post-error adaptations influence distractor processing, but also that the BPL signal is in general sensitive to the degree of response conflict induced per trial by comparing close and far distractor trials (Supplementary Figure 5).

Shade = 99% CI, grey background = significant time points between conditions after Bonferroni correction. Vertical coloured lines = flanker / response onset per category as in Figure 6.

2) Regarding evidence accumulation in the beta signal, the authors interpret this as a clear state of evidence accumulation (p.g 5), yet this is highly problematic. I accept the argument that motor cortex could accumulate evidence. Yet other brain areas may be involved as well (e.g. the authors acknowledge the role of P3 in this process in the discussion). The motor cortex activity may result from the output of a separate accumulator, making it definitively epiphenomenal. This possibility is not adequately addressed, and it strongly affects the confidence by which the authors can assume that latent computational features map onto manifest indicators. The authors seem tangentially aware of this possibility (“pg. 10: regions deemed critical for decision formation may rather be epiphenomenal”), but do not apply this intellectual rigor to their own assumptions.

As has also been pointed out by Reviewer 1, for the current study it is not relevant whether beta power is causally related to decision formation. In fact, our experiment does not allow any such conclusions. However, we provide evidence that BPL can be used as a read-out of decision formation, making it a useful tool to study decision-making even if future studies may find, using interventional approaches, that beta power is epiphenomenal. Additionally, we are not aware of any decision-correlate that can be measured using neurophysiological techniques and has yet been successfully manipulated in order to influence decision formation. We have removed the discussion of whether BPL is epiphenomenal or not from the manuscript, as was also suggested by Reviewer #1.

With regard to the scope of the paper, please also see our response to point 3) below.

3) In line with the concern above, the focus of the report solely on motor activities is too limited for the aims that the authors are trying to address. The role of P3 or other accumulators is one example. Yet on pg. 7 the authors aim to address “more evidence for increased attentional control”, which may be most face-validly addressed via visual cortex activity (such as alpha band suppression which is known to be altered after errors, see Rebecca Compton’s work). The authors even suppose on pg. 9 that visual processing changes following errors (“focused on relevant positions following an error”), yet they follow it up with motor cortex signals.

We agree with the reviewer that it would be nice to directly study post-error changes in selective attention as well. However, our task is not designed and not well-suited to study subtle changes at the perceptual level which might reflect adaptations in selective attention. Indeed, we are following up these questions in different paradigms. Studying alpha power as a measure of selective attention would require two conditions to be met: (1) sufficiently long intervals between relevant events to enable unequivocal attribution of the finding to stimulus processing and (2) that targets and distractors are represented in spatially distinct regions of visual cortex, ideally in different hemispheres. The fast timing of our task and the nature of stimuli in flanker tasks do not fit these conditions. Therefore, we have decided not to investigate changes in alpha activity. Studying VEPs to capture attentional control has been attempted by a number of colleagues and ourselves in multiple tasks. Regrettably, most of these attempts we are aware of, did not yield clear results, often no significant change was found.

Publication bias likely obscures this as these null findings have not been published. Exceptions were published by Buzzell et al., 2017, Nigbur et al., 2015, and Steinhauser et al., 2017. Only Nigbur and colleagues used a flanker task, but notably the VEPs were elicited by additional probe stimuli. Another study investigating cognitive-control-related modulations of stimulus-locked ERPs in a flanker task (Wendt et al., 2009) did only report changes in the N2 but not in earlier VEPs.

Nevertheless, we performed an analysis of VEPs following errors and plotted results below. Due to the short RSIs in our task, we separated the analysis additionally by short (250 ms) and long (700 ms) RSIs to test if the effects were induced by overlap of error-related EEG activity. Baseline correction was performed from -100 to 0 ms around stimulus onset, to maximize the distance from the last response (but see below). We plot VEPs of current source density transformed signals to further reduce possible overlap with frontal components in Figure Response 2. We additionally ran a single-trial regression onto the whole set of scalp electrodes according to the following equation:

$$(R1) \text{CSD} - \text{EEG} = b_0 + b_1 \times \text{Accuracy} + b_2 \times \text{Incongruence} + b_3 \times \log - RT + b_4 \times \text{Previous Accuracy} + b_5 \times \text{RSI} + b_6 \times \text{Distance} + e.$$

We additionally included an interaction for *Incongruence* \times *Previous Accuracy* to test if a possible main effect of *Incongruence* in visual areas would be influenced by previous accuracy. While we did find evidence for VEP modulation by stimulus *Incongruence* (Fig. Resp. 2k), this does not interact with the accuracy of the previous trial (Fig. Resp. 2l). As for a main effect of *Previous Accuracy*, we do not find convincing evidence for VEP changes following errors. The regression analysis indicated that there is some change at lateral occipital electrodes (Fig. Resp. 2e), but this depended on whether all trials (top rows) or only long RSI trials were included into the analysis. The effect was additionally not focused on a distinct visual component, but rather long-lasting, suggestive of baseline confounds. Therefore, we do not consider this general effect a clear representation of changed attentional focus.

Some things should be considered. Our task is not designed for such analyses: We used an SOI between flanker and target onset of 83 ms. Therefore, VEPs are overlapping and specifically the N1 component (which usually peaks around 200 ms) induced by flankers is overlapped by the P1 (which usually peaks around 130 ms) induced by the target. Furthermore, because we used different distances between flanker and target stimulus, to modulate response conflict depending on visual proximity, and because VEPs are sensitive to this modulation to some degree (Fig. Resp. 2m), there may be additional variance in the VEPs which complicates the analyses of post-error trials even further. However, this finding may still indicate that changes in flanker suppression which are clearly evident in both the lateralised beta-signal and its temporal derivative, and which we confirm using the DDM on a behavioural level which predicts very similar changes in the computation-time-course of the decision variable, happen at later stages when visual information is integrated into choice selection. Note that the BPL signal by using differences between both hemispheres is not affected by baseline changes and no post-error effects are seen in the pre-stimulus phase when BPL is analysed.

Figure Response 2: Analysis of VEPs in CSD-transformed EEG signals split by all trials (top row) and long-RSI (bottom row). There were no consistent patterns of changes induced by previous accuracy present in VEPs. Additionally, component overlap of the previous trial was present in the EEG components revealed in the topography plots (e,j). Additionally, VEPs were sensitive to the *Incongruence* (k) of a given trial, but this effect was not modulated by whether or not the previous trial was an error (l). Finally, the distance between flanker and target stimulus (k) modulated the signal and more negative P1 amplitudes were observed in the close condition. Non-significant points in the topography plots are masked out in white at a lenient threshold ($p = 0.001$). Shades in the time-course plots = 99.9% CI.

We feel that the null-finding cannot be interpreted as evidence against possible post-error attentional control. It appears that flanker tasks with a stimulus-onset asynchrony between flankers and target, VEPs are not well suited and have too low sensitivity. Moreover, while our DDM and BPL data suggest suppression of flanker-induced evidence, this suppression could be implemented at several levels of information processing and does not need to be confined to attentional regulation alone (see also our response to Reviewer #1, point 7).

We feel that the manuscript, as well as the supplementary material, already contains many complex analyses and that the VEP results would add a different layer to the context of the manuscript than is intended by us. Therefore, we have not included these results into the current manuscript. If the reviewer feels that we should add these analyses, we are happy to include them into the supplementary material as well.

4) I'm not convinced that these latent constructs can be firmly tested without application of a drift diffusion model (DDM), which somewhat tacitly determines the objective definition of these constructs.

In the revised version of the manuscript, we have constructed a DDM that allows to simulate the current task and relevant aspects have been reproduced in response to point 1) above.

Notably, we do find remarkable similarities between BPL and both the time-course of the DDM decision variable, as well as its properties in predicting response speed and accuracy biased by pre-stimulus state (start point variance, sz), stimulus congruence, and post-error adaptation effects. We would like to point out again, that our DDM was not informed by the neural signal. In fact, we took great care in avoiding circularity as far as possible and only used very sparse assumptions to design the DDM (existence of pre-stimulus effects in the form of trial-wise start-point biases, the possibility to suppress flanker effects and that flanker suppression is variable over trials).

a) Pg. 7: "The orienting account show be associated with general slowing of evidence accumulation". Is this necessarily a prediction of the orienting account? I think it could certainly predict an increase in non-decision time alone, which would be different than a threshold increase that would allow increased accumulation. This would have different implications for downstream processes, and it could certainly be a feature within the adaptive account. Without a DDM, this possibility cannot be tested.

We thank the reviewer for pointing out that the orienting account could additionally predict increased non-decision time (T_{er}). We added a specific comparison of DDM parameters that were fit to post-error and post-correct trials alone. The result clearly demonstrates changes in boundary, drift-rate and the degree to which flankers can be suppressed in accordance with focussed attention. As to the point above, despite around 850 participants and individually fitting parameters to each subject, we find no evidence of a change in non-decision time following errors ($p = 0.1$, uncorrected). Overall, the suggested addition of the formal DDM has considerably strengthened the manuscript.

b) I was not fully convinced that the slope of beta power reflects evidence accumulation, epiphenomenon or not. There simply wasn't enough information and tests of this hypothesis for me to confidently assume this process was occurring during this aspect of the signal. This is critical, as strong theoretical interpretations are made on pg. 9 of the discussion based on this tenuous operationalization ("The slope of BLP indicates the net speed of evidence accumulation in favor of one response"; "this non difference in evidence accumulation speed favors adative error processing accounts ...). DDM would help to bolster this assumption.

We have added a DDM as described above. Furthermore, we have added more detailed analyses of beta slopes and we display that this measure is sensitive to cognitive control effects during flanker processing (it reflects increases in flanker processing when response conflict is high in the task), speed of responding and post-error adaptations (PERI). The relevant sections are reproduced below for the reviewer's convenience:

Figure 8. Beta slope dissociates specific from general adaptations.

BPL slope indexes the steepness of the BPL signal analysed via single-trial fitting of individual periods in the BPL signal surrounding every data point (80 ms). The slope can be seen as the rate of change in BPL, thus reflecting how quickly beta power lateralises to one hemisphere. (a) displays how BPL slopes differ between congruent (blue) and incongruent (orange) trials. The dashed vertical line reflects when BPL slopes start to first significantly deviate from zero, which coincides with the non-decision time parameter in the DDM.

(b) displays that the degree of interference induced by visual proximity of the distractors affects the duration of BPL slopes that point into the incorrect (distractor-) direction. As a result, when incongruent flankers are close to the target, BPL to the wrong direction is stronger and longer-lasting as compared to far flanker trials (cf. Fig. S6). Similarly, for incongruent trials the slope of distractor processing is influenced by previous accuracy (c), which is consistent with PERI. However, there is no significant difference (even at a very lenient threshold of $p < 0.05$) at any point during early BPL slopes when comparing congruent post-error (orange) to post-correct (blue) trials. This suggests that post-error trials specifically reduce the interference induced by distractors, which is seen in the BPL slope analysis at around 290 to 360 ms.

For an additional analysis comparing how slow and fast responses are associated with BPL slope, see Supplementary Fig. 7. Statistical comparisons between conditions are calculated via repeated t -tests corrected for multiple comparisons (a,b) and uncorrected (c,d) and significant time periods are shaded in grey, shades reflect SE.

Figure S8. BPL slope for fast and slow trials.

Akin to main Figure 8, that displays stimulus processing for trials separated by conflict, degree of interference and post-error effects, we display here how response speed is reflected in the BPL slope signal. On congruent (a) as well as incongruent (b) trials, slower responses are preceded by very early slopes hinting in the direction of the response that will not be selected later in the trial (vertical lines represent mean RT per condition). Slope changes induced by incongruent trials (b) are comparable during the time distractors are processed, but more quickly gain in steepness on fast trials (effect 340 to 380 ms). Interestingly, on congruent trials BPL slopes have returned to baseline when responses are made, whereas they continue to unfold in the incongruent condition.

Grey areas reflect significant time-points of paired t -test between conditions following Bonferroni correction, shades reflect SE.

c) In general, the effects here are tested as being linear expect in the case of statistical moderation. Pg. 5: "Thus, behavioral slowing induced by errors mediates no further accuracy increase as would be expected on comparatively slow trials that were preceded by correct responses". I'm not sure that post-error responses don't contain a meaningful degree of non-linearity compared to post-correct trials. As shown by Kiani & Purcell (2016), accumulation and threshold can interact in complex ways following errors; the addition of an orienting response could interact with adaptive process in a variety of manner (see 4a above). I think including a formal DDM account of these data would be one positive step towards being able to make the conclusions that the authors are trying to make here.

Thank you for pointing this out. Indeed, the assumption of a linear relationship between accuracy and RT was oversimplified in our previous analysis and led us to the conclusion that PES can fully explain accuracy changes following error responses. We have added the DDM analyses (as pointed out above) and have removed this regression analysis from the manuscript as we feel it does not provide any additional information beyond what can be seen from the result of the DDM. We feel that this has helped to focus the manuscript more clearly onto relevant points.

Reviewer #3 (Remarks to the Author):

This EEG study investigates the neural mechanisms of post-error adaptation. The study addresses a timely and relevant question. The applied methods are sound, the paper is based on an exceptionally large dataset, is generally well written and presents several interesting findings. Nevertheless, I feel that the conceptual advance of the paper is limited by several shortcomings.

We are pleased with the reviewer's assessment that our study is timely and that the question is relevant. We have added several additional analyses including a full DDM, simulations of the decision variable based on the DDM that was purely fit to behavioural data, yet surprisingly well matches the BPL signal, and a more precise analysis of the contributions of individual hemispheres to the BPL signal. Notably, we are able to capture cognitive control effects within the DDM and compare these directly to BPL. We are not aware of a study that would have demonstrated such a strong link between the DDM and an independent neural signal, thus we feel that our results, apart from tracing apart the constituents of post-error adaptive processes, contributes a significant conceptual advance.

1) I feel that previous studies that already showed key results of the present study are not sufficiently discussed. This includes in particular the general role of beta-lateralization as a marker of decision-dynamics and the post-error threshold-increase. In particular, Purcell et al. *Neuron*, 2016 presented not-only strong neurophysiological evidence for a post-error threshold-increase, but also converging evidence for this increase based on computational modeling of behavior (drift-diffusion-model, DDM). Similarly, Dutilh et al, 2012 (DOI 10.3758/s13414-011-0243-2) already presented strong DDM-based evidence for a post-error threshold-increase. In light of these findings, the reported post-error threshold-increase seems somewhat confirmatory.

We have extended the discussion of the Purcell & Kiani paper and included the Dutilh et al. paper as well. Furthermore, we now include a detailed DDM approach of not only the behaviour, but also of the simulated time-course of the decision variable derived from the DDM. This simulation displays striking similarities with multiple aspects seen in BPL, for example prior to stimulus presentation in the form of start-point variance in the DDM that is qualitatively reflected in the BPL signal. We additionally confirm with the DDM that boundaries following errors increase (as we had reported compatible with the BPL findings before). Furthermore, we show selective flanker suppression both in an additional DDM parameter that is fit to all participants and again is compatible with the BPL signal.

We agree that valuable previous work in the post-error adaptations already exists. However, we think that our study by now combining a formal computational model with a highly time-resolved neural measure of decision formation in humans, as well as by making use of an exceptionally powerful dataset, provides a strong contribution to the field.

We reproduce the most relevant aspects of DDM predictions in relation to the BPL signal below for the reviewer's convenience:

Figure S1. Illustration of the multi-stage DDM.

Stimuli in the arrow flanker task (a). At t_0 flankers were presented either closer (second screen) or further away (see Fig. 1) from the delayed (stimulus-onset-asynchrony (SOA) = 83 ms) target (third screen). The relative direction between flankers

and target determines a trials' congruence. Only the target arrow indicates the correct response (green hand = correct congruent, orange = correct incongruent, red = error incongruent). These task phases are explicitly modelled in the DDM (b) with 5 separate stages. This was done as to model the task as closely as possible.

Stage 1 consists of a pre-stimulus baseline for which the value is determined by the start-point variance modelled as a uniform distribution. We did not assume general bias in start points.

Stage 2 models the non-decision time per trial that varies according to the variance parameter st again assuming a uniform distribution ($Ter_{(t)} = Ter \pm st \times 0.5$).

Stage 3 represents a noisy diffusion in flanker direction for the duration of the SOA. On incongruent trials (orange), the decision process drifts away from the correct decision with drift rate $v_{(t)} \times f_{(t)}$. The trialwise variance parameters sv and sf were modelled as Gaussian distributions with mean v and f respectively. Note that lower values indicate stronger suppression of distractors and $f = 1$ indicates no flanker suppression, whereas values > 1 would indicate stronger flanker compared to target processing.

Stage 4 models the consecutive diffusion back into the correct (left hand) direction with drift rate $v_{(t)}$. Because on incongruent trials the drift in the SOA was opposite to the consecutive drift in stage 4, RTs are longer. Error likelihood increases when the baseline is shifted towards the flanker direction and $f_{(t)}$ is higher on a given trial resulting in an early crossing of decision boundaries (red line). Note that this implies that errors are usually faster on incongruent trials.

Stage 5 models the return to baseline of the decision variable as a half-normal distribution with variance 0.1 and start point 1 that downscaled the drift signal once a decision was made, akin to an Ornstein-Uhlenbeck process. This was done as it is unclear if under speeded conditions a decision process truly terminates when a response occurs or continues to evaluate evidence accumulation for a certain period of time to possibly allow quick adjustments. However, this has no influence on model predictions regarding RT and accuracy and only determines how the diffusion signal behaves *after* a decision has been made which does not influence the start of the next trial.

Grey shades illustrate the distribution of trial-by-trial variance parameters (sz = start point variability, sv = drift rate variability, sf = flanker suppression variability, st = non-decision time variability). $\pm a$ = upper / lower decision boundary, $z_{(t)}$ = start-point per trial, $v_{(t)}$ = drift rate per trial, $Ter_{(t)}$ = non-decision time per trial, $f_{(t)}$ = speed of flanker relative to target diffusion per trial. $R_{err,con,inc}$ = time of error, congruent and incongruent response.

An extended DDM captures task effects. Simple accumulator models integrate sensory evidence into a decision variable that determines choices and RT. This process has been found to be reflected in single neurons in various cortical areas as revealed by intracranial recordings in monkeys^{42,43}. Many of the recorded neurons reflected the accumulated evidence and triggered a response when a common threshold was reached. A simple variant of these accumulator models is the DDM which assumes that choice options are mutually exclusive. This allows to replace two separate accumulators with one decision variable that reflects the difference between both decision options. On a given trial, the decision can be randomly biased to favour one response (*start-point variance*). Because decisions are triggered when a decision boundary is crossed, the height of the boundary parameter (a) determines how much integration of evidence is required to trigger a response. Finally, visual processing and motor output is integrated in the model by a parameter that reflects the non-decision time (Ter), which in our model can vary from trial to trial (st).

Usually, DDMs assume a constant speed of evidence accumulation within each trial, which is governed by a drift rate parameter (v , Fig. 3a) and trialwise variance (sv). However, in conflict tasks such as the flanker paradigm, evidence can first point into one direction and thereafter reverse. To reflect this, we used a multi-stage DDM. In our model, evidence accumulation followed the flankers' direction during the time they were displayed on screen alone (83 ms). After target onset, evidence accumulation was driven by the direction of the target.

We fit three variants of the model to the distribution of RTs for each participant using QMLE⁴⁴ and differential evolution⁴⁵. The first model was a standard DDM that used the same evidence accumulation rate for distractor and target stimuli. The second DDM included the possibility to relatively downweigh, i.e., suppress distractor evidence (parameter f), and the third model additionally allowed trial-by-trial variance in distractor suppression (sf). Model comparison (Fig. 3b) suggested that the last model provided the best fit to the data indicating that participants suppress distractor information, yet this varies between individual trials. Suppression (rather than amplification) of distractors was confirmed by the value of f , which was significantly lower than 1, where 1 would reflect equal processing of distractor and target information (mean $f = 0.44 \pm 0.03$ (99.9% CI), $t_{862} = 64.3$, $p = 0$ within precision). Overall, the model provided a good fit to the data and matched participants RTs on congruent, incongruent and error trials as well as accuracies in congruent and incongruent trials (Fig. 3c-f).

Association between model parameters and behaviour. We then tested which model parameters were associated most strongly across the group of participants with RT, accuracy, interference and the ratio of errors in congruent relative to incongruent trials (Fig. 3g-m). This revealed that RT was mainly reflected by the drift rate parameter (v) and accuracy by the height of the boundary (a) and strength (f) and variance (sf) of distractor suppression. These latter two parameters additionally covaried with the magnitude of the interference effect on incongruent trials per participant (Fig. 3l) as well as with the ratio of incongruent to congruent errors (Fig. 3m). We use this model to provide a quantitative analysis of post-error effects and make predictions on how such a decision variable should behave when it is analysed for task effects. Notably, the model also predicted errors on congruent trials (Fig. 4c), although these were removed for fitting (see Supplements for details).

Finally, we investigated which variance parameters on a single-trial basis affect the models' accuracy and RT using a similar regression approach as in the behavioural analysis (GLM 3, Methods). We found that both accuracy ($p < 10^{-10}$) and RT ($p < 10^{-33}$) were significantly reduced when a trial was biased towards the response that was later on selected due to variance in start-points (sz). Additionally, the degree of trial-wise suppression of distractor information (determined by factor sf in the

model) was strongly associated with accuracy ($p < 10^{-52}$) and less with RT ($p < 10^{-9}$). This analysis identifies that especially variance in start-point and flanker suppression are factors that drive individual trials in the model to be fast or slow and correct or incorrect.

Post-error adaptations in the DDM. Next, we fit the same DDM as before to post-error and post-correct trials separately. We excluded participants with less than 30 valid post-error trials ($n = 15$). Additionally, we fixed the variance parameters of the DDM (sv , st , $StarVar$, sf) to the group mean to facilitate convergence. Note that variance parameters were still in the model, only their value did not vary between participants. The DDM captured RT and accuracy well in both post-error and post-correct trials (Fig. 4a-b, d-e).

Next, we compared differences in parameter values for drift rate, boundary, non-decision time and flanker suppression in post-error and post-correct model fits using multiple logistic regression of parameter values onto which trials' the model was fit to according to eq. 5 (Methods). Positive regression coefficients indicate parameter increases following errors (Fig. 5). We found that drift-rate was decreased ($t_{1691} = -15.3$, $p < 10^{-51}$), boundaries increased ($t_{1691} = 17.5$, $p < 10^{-66}$), and non-decision time unchanged following errors ($t_{1691} = -1.7$, $p = 0.1$). Additionally, flankers were more strongly suppressed ($t_{1691} = -5.3$, $p < 10^{-6}$) in post-error trials. This suggests that slower evidence accumulation began at the same time (no change in Ter) and that distractors were more strongly suppressed following errors. A separate parameter recovery analysis confirmed that our fit method reliably identified model parameters (Supplementary Figure 3).

Figure 3. Drift diffusion model fits and parameters.

Our implementation of the drift diffusion model (DDM) included 8 free parameters (a). Apart from the standard parameters coding for drift rate (v), boundary (a), and non-decision time (Ter), we included variance parameters that determined how much these three parameters varied from trial-to-trial (see Supplementary Fig. 1 for more details about the model). Model comparisons (b), revealed that two additional free parameters for distractor suppression (red) and trialwise variance of distractor suppression (yellow) increased model fit (see Supplementary Fig. S2 for results of worse fitting models). The flanker suppression parameter f modulates flanker processing and scales the drift rate during the time the distractors are on screen. Additionally, Gaussian variance (sf) for this parameter that influenced trial-wise values, further increased model fit (b). (c-e) shows quantile fits of the model (dark blue) against human RT data and (f) shows model and human accuracy. In all conditions (congruent & incongruent correct as well as incongruent error), the model captures the RT data in each quantile, suggesting a good fit to the data. Boxes = interquartile range (IQR), o = median, - = mean, whiskers = 1.5 x IQR, grey dots = outlier. Note that we removed congruent errors from the analysis as these were very rare (<2.5% of trials) and some subjects did not commit congruent errors at all. (g-m) displays the relationship between model parameters and behaviour across subjects. Displayed are regression coefficients and 99.9% confidence intervals.

(g) Faster participants were fit by higher drift rates, lower decision boundaries, and lower non-decision times.

(h-j) RT on error trials (h) was most strongly dependent on the non-decision time because errors are usually very fast, whereas drift rates are more closely associated with RT on correct congruent (i) and incongruent (j) trials.

(k) Accuracy was most strongly reflected in the height of the boundary (a) but additionally dependent on how much distractors were suppressed (f) and how variable this suppression from trial-to-trial was (sf). Higher variance reduces accuracy, because in more trials distractors are likely to not be suppressed.

(l) Interference and the ratio of errors between incongruent and congruent trials (m) additionally covaried with distractor suppression and its variance (f , sf) and the less flankers were suppressed in the model, the higher was the interference and the more incongruent compared to congruent errors a participant made.

Figure 4. How model parameters influence accuracy and RT.

Displayed are regression coefficients of a single-trial regression analysis within the DDM using trial-wise variance parameters. The goal of this analysis was to delineate which parameters have strong impact on whether a trial will be correct (a) or fast (b). Akin to the participant data (Fig. 2), we used logistic and multivariate regression to investigate accuracy and RT, respectively. In both models, a trial's incongruence had a strong impact (yellow, compare to Fig. 2a,e). Variance in the start-point of the decision process of each trial did not overall influence accuracy and RT. However, how much the start-point was biased towards the response the model later on selected, significantly decreased accuracy and RT (factor *SP Resp*). This effect was less pronounced for the absolute bias per trial (factor *SP Abs*). Trialwise faster drift rates significantly decreased RT and much less so accuracy. When flankers were more strongly suppressed (f is lower), accuracy increased and RT slightly decreased. Non-decision time (T_{er}) did not influence accuracy, but longer T_{er} on each single trial led to increased RT. Error bars reflect 99.9% CI, plotted are regression weights for one DDM that employed the group mean parameters for all post-correct trials and simulation of 5.000 single trials.

Figure 5. Model changes following errors.

When we fit the model separately to all trials following correct responses (a,b) and all trials following errors (d,e), we again observed that the model captured RT distributions in all quantiles of the data despite partly limited trials in the post-error condition. By comparing (a) against (d) and (b) against (e) it can also be seen, that the model captured PES. Furthermore, the model displayed significant accuracy increases especially in incongruent trials (c) post-error (t -test $t_{348} = 17.5$, $p < 10^{-56}$).

(f) Model parameter values were compared using logistic regression onto *TrialType* (post-error, post-correct) according to eq. 5. Plotted are regression weights and standard errors for individual factor contributions to the difference between models fitted to post-error and post-correct trials and positive values indicate parameter increases following errors. The largest change in parameter values was observed for decision boundaries, which we found to be increased when the model was fit to post-error trials. Additionally, drift rates were decreased, yet the non-decision time was unchanged. Crucially, the degree to which distractors influenced the decision variable (multiple linear regression $t_{1691} = -5.3$, $p = 1.24 \times 10^{-7}$) was decreased following errors, which was seen over and above decreased drift rates or increased boundaries.

Plot conventions for a-e as in Fig. 3, error-bars in f reflect 99.9% CI.

Figure 6. Stimulus related beta power and simulated decision variable.

(a) The BPL is composed of the activity difference recorded over contralateral (green) and ipsilateral motor cortices (blue). A statistically significant lateralisation towards the contralateral motor cortex is seen around 200 ms prior to response execution (b). The inset shows a regression weight map of the scalp topography of beta power when regressed against the side of motor execution according to GLM 3 (see Methods and Supplementary Fig. 4 for details).

(c,e) Comparison of distractor effects on BPL (c) and modelled decision signal (e) in the DDM during stimulus processing. Distractor effects were seen at very similar times on incongruent (red) compared to congruent trials (blue): distractors pushed BPL and the decision variable away from the correct response. (d) Across participants, the degree of distractor representation in BPL (mean between 340 – 420 ms) positively correlated ($r = .20$, $p < 10^{-16}$) with the interference effect on RT.

Notably, many features of the time-course and dynamics of BPL were closely predicted by the diffusion variable of the DDM. Non-decision time (Ter) reflects stimulus processing before it influences the decision in the DDM and the onset of BPL matches Ter plus its variance (dots surrounding vertical dashed line in c and e). The similarities extend to pre-stimulus effects. Here, both signals are lateralised towards the chosen response which is caused by single-trial start-point variance in the DDM affecting accuracy and RT. Strong pre-stimulus bias leads to errors (g) but also quick (based on median splitting) responses (i) in the model. The same pattern is evident in the BPL plots (c,f,h), suggesting that BPL influences speed and accuracy of upcoming responses. In the DDM, this is caused by the causal relationship between start-point variance and accuracy and RT. The striking similarity between diffusion signal and BPL hints at similar relationships for the neural data. Note that the pre-stimulus bias is absent when all trials are plotted without mapping to the executed response.

Shades represent 99.9% CI, the vertical blue and orange lines in c-i reflect the mean RT per condition in humans (top) and model (bottom). Grey background in c,f,h indicate significant time-points between conditions after Bonferroni correction. For details on how the simulation the DDM was performed, see Methods, Supplemental Material and Supplemental Figure 1.

Figure 7. Post-error adaptations in beta power and simulated choices.

Plotted are BPL time-courses following error and correct responses for all trials (a,b) and incongruent trials specifically (e,d). In the lower row the simulated decision signal derived from the DDMs fit to post-correct and post-error trials is plotted split up by the same factors (e-h). We found evidence for increased BPL at response execution (b), which in the DDM is caused by boundary increases following errors (f). Additionally, post-error reduction of interference (PERI) is seen in the beta signal (c,d) and predicted by the DDM decision variable as well (g,h). We additionally confirmed that not only post-error adaptations influence distractor processing, but also that the BPL signal is in general sensitive to the degree of response conflict induced per trial by comparing close and far distractor trials (Supplementary Figure 5). Shade = 99% CI, grey background = significant time points between conditions after Bonferroni correction. Vertical coloured lines = flanker / response onset per category as in Figure 6.

2) I feel that the most interesting finding of the present study is the claimed absence of a post-error decrease of target processing, which is interpreted as a post-error focusing of attention. This is in contrast to several previous reports (e.g., Purcell et al. 2016). However, this central conclusion is based merely on a negative finding, i.e. the absence of a slope-effect of preceding errors. Thus, this claim should be substantiated by further evidence.

E.g., I miss a comparison of target-evoked responses (ERFs) following errors or correct trials, which could reveal differences in target processing. Furthermore, I miss a computational model of the behavioral data (DDM), which could provide complementary evidence for the authors' conclusions.

Independent from these additional analyses, the key result (slope fit) needs to be better presented. E.g., the fitted linear models should be shown with the measured data in order to assess the goodness-of-fit and to show how constrained these fits are. In fact, from Fig. 4g, I feel that the post-error data is rather noisy and that the slope could well be shallower for post-error as compared to post-correct trials.

Thank you for this valid point. First, we would like to remark that our findings are not strictly in contrast with the results of Purcell & Kiani, which was also outlined by reviewer #1 (point 7), because in their task attentional focussing is not an actual strategy as better visual discrimination would require a reduction of processing noise. This is arguably much less feasible than focussing attention to a known target on a screen. Thus, adaptations in our task can be orchestrated prior to the next trial and specifically adjusted to the stimulus set whereas such changes in dot-motion tasks would reflect very broad attentional aspects or require increases in signal-to-noise of the visual system, which may not be possible if participants attentively process that task. We discuss this now in more detail in the revised version of the manuscript.

We agree that the selective reduction of target processing on incongruent trials is an essential finding of our study. However, this conclusion is not purely based on the absence of a difference in slopes on congruent-post error trials. We additionally show (Fig. R7) that distractors drive the BPL signal in their direction less strongly following errors. This change was predicted by the DDM fit to post-error and post-correct trials and the temporal evolutions from the reconstructed decision variable is strikingly similar to the recorded beta signal (compare Fig. R7c,g reproduced above). Furthermore, we now also display that post-error effects can be seen on incongruent trials reflected in steeper slopes. Therefore, we demonstrate the BPL slopes in general are sensitive to evidence accumulation rate changes, which we confirm by comparing the speed of responses as well as conflict processing (plots reproduced below). Thus, the absence of an effect on congruent post-error trials is at least suggestive of a selective effect.

We furthermore agree that the slope analyses were unsatisfying in the previous version of the paper. We have induced several changes to them. In order to increase signal-to-noise ratio, we used a broader time window for the slope fit (80 ms instead of 20 ms). Additionally, we increased the temporal resolution of the analysis. This resulted in much clearer results

with a better SNR. Finally, our DDM analysis indicated additional flanker suppression following error trials, corroborating our findings from a computational modelling perspective as well. We have reproduced the corresponding Figure and relevant passages of the revised manuscript below.

Figure 8. Beta slope dissociates specific from general adaptations.

BPL slope indexes the steepness of the BPL signal analysed via single-trial fitting of individual periods in the BPL signal surrounding every data point (80 ms). The slope can be seen as the rate of change in BPL, thus reflecting how quickly beta power lateralises to one hemisphere. (a) displays how BPL slopes differ between congruent (blue) and incongruent (orange) trials. The dashed vertical line reflects when BPL slopes start to first significantly deviate from zero, which coincides with the non-decision time parameter in the DDM.

(b) displays that the degree of interference induced by visual proximity of the distractors affects the duration of BPL slopes that point into the incorrect (distractor-) direction. As a result, when incongruent flankers are close to the target, BPL to the wrong direction is stronger and longer-lasting as compared to far flanker trials (cf. Fig. S6). Similarly, for incongruent trials the slope of distractor processing is influenced by previous accuracy (c), which is consistent with PERI. However, there is no significant difference (even at a very lenient threshold of $p < 0.05$) at any point during early BPL slopes when comparing congruent post-error (orange) to post-correct (blue) trials. This suggests that post-error trials specifically reduce the interference induced by distractors, which is seen in the BPL slope analysis at around 290 to 360 ms.

For an additional analysis comparing how slow and fast responses are associated with BPL slope, see Supplementary Fig. 7. Statistical comparisons between conditions are calculated via repeated t -tests corrected for multiple comparisons (a,b) and uncorrected (c,d) and significant time periods are shaded in grey, shades reflect SE.

Figure S8. BPL slope for fast and slow trials.

Akin to main Figure 8, that displays stimulus processing for trials separated by conflict, degree of interference and post-error effects, we display here how response speed is reflected in the BPL slope signal. On congruent (a) as well as incongruent (b) trials, slower responses are preceded by very early slopes hinting in the direction of the response that will not be selected

later in the trial (vertical lines represent mean RT per condition). Slope changes induced by incongruent trials (b) are comparable during the time distractors are processed, but more quickly gain in steepness on fast trials (effect 340 to 380 ms). Interestingly, on congruent trials BPL slopes have returned to baseline when responses are made, whereas they continue to unfold in the incongruent condition. Grey areas reflect significant time-points of paired *t*-test between conditions following Bonferroni correction, shades reflect SE.

3) The stronger BPL for congruent as compared to incongruent trials (Fig. 3g) is unexpected and casts doubt on the BPL at response-time as a reliable indicator of decision-threshold. This needs to be discussed.

We have further dissected BPL at response times into the constituent motor-cortex contributions which we did as well for the pre-stimulus time period. The latter analysis clearly suggested that BPL contains information that cannot be found when only one hemisphere is analysed, therefore demonstrating that the relative degree of engagement of either hemisphere is a relevant factor for response preparation. We demonstrate that BPL even before stimulus onset influences speed and accuracy of upcoming choices, closely resembling start-point variance in the DDM framework. Such characteristics are neither matched by the ipsi- nor the contralateral hemisphere beta signals.

With regard to response-related influences on BPL, we found that the effect induced by incongruent stimuli is due to reduced ipsilateral BP decreases rather than a threshold reduction on the contralateral hemisphere. Therefore, clearly there are multiple influences on BPL which cannot easily be captured by one interpretation. We acknowledge this and discuss accounts of what is represented in the BPL signal now in more detail.

We reproduce the revised Figures entailing the single-trial regression on BPL at response (now Figure R9) and Supplementary Figure S5 below.

Figure 9. Single-trial regression on beta at response.

a) Results of a single-trial regression analysis on BPL at response (mean \pm 12 ms surrounding button press), comparing relevant task factors. This analysis was performed to rule out that the results demonstrated for RT and post-error effects result from averaging over trials. (b-d) reflect raw values splits for *Incongruence* (congruent vs incongruent), *RSI* (short vs fast) and *Previous Accuracy*

(correct vs error). (f-h) display results of separate regression analyses split by ipsilateral (hemisphere that does not induce the choice on the current trial) and contralateral hemispheres.

Following errors, BPL is larger (more negative) which is caused by a stronger increase in BP over the ipsilateral hemisphere, suggesting that following errors, the unchosen response is more effectively suppressed. Note that in general (Fig. 6a), beta power strongly decreases before responses are executed. When responses are slow, BPL is decreased. This is partly due to less decreases over the contralateral hemisphere, but the effect is much stronger when the relative signal per trial is investigated than when either hemisphere on its own is analysed (h). Furthermore, incongruent trials cause reduced BPL. This latter result is induced by changes over the ipsilateral hemisphere (f) where BP more strongly reduced on incongruent trials, which could reflect the preactivation of the competing response tendency.

Error-bars reflect 99.9% CI, a) displays mean within participants t-values, and statistics are results of t-tests of individual within-subject regressions against zero. RT in (e) is trichotomised. See Supplementary Figure 5 for a similar analysis on pre-stimulus BP. NLD reflects the distance from the last break in the task, and trial number the general time on task.

Figure S5. Single-trial regression on pre-stimulus beta power.

Because we found that the DDM decision variable predicted plausible influences of the state of the decision in the baseline period and during the random noise that was not driven by any stimulus input, we looked more specifically for ipsilateral, contralateral and BPL associations before stimulus onset (-100 until 0 ms around flanker onset). We used the same regression as in Fig. 9 extended by the current trials (future) accuracy. For post-error effects, we found that BPL was unchanged ($p = 1$ corrected), whereas beta power over ipsi- ($t_{862} = 15.9, p < 10^{-48}$) and contralateral hemispheres ($t_{862} = 15.1, p < 10^{-44}$) was

strongly increased following errors which demonstrates that global, yet not BPL unspecifically increases following errors. Interestingly, while ipsilateral BP ($t_{862} = 2.1$, $p = 0.25$) was not significantly related to RT, contralateral BP ($t_{862} = 11.9$, $p < 10^{-28}$) showed positive associations with RT (stronger BP was followed by higher RT), BPL was most strongly associated with the RT of the upcoming choice ($t_{862} = 19.2$, $p < 10^{-66}$). Moreover, while neither ipsi- ($p = 1$) nor contralateral ($p = .15$) BP were predictive of a trials' accuracy, BPL was ($t_{862} = -6$, $p < 10^{-23}$). Note that the significant effect of *Incongruence* does not indicate a spurious result but is a result of mapping data to the given response. It indicates that even before stimulus onset the association between the later selected response and BPL is stronger in incongruent compared to congruent trials ($t_{862} = -15.6$, $p < 10^{-47}$). The same is seen in the DDM, where start-point variability influences decisions more strongly when the trial is incongruent as the decision boundary is reached early during the time of flanker processing. These results demonstrate that BPL is a measure that cannot be reduced to the simple analysis of both hemispheres alone because neither was associated with upcoming accuracy.

All follow-up plots show raw BP / BPL data (insets on the right), error-bars reflect 99% CI for regression weights, and 99.99% CI for raw data. Results are adjusted for multiple comparisons via Bonferroni correction. NLD reflects the log-distance from the last break in the task, and trial number the general time on task.

References

- Danielmeier, C., & Ullsperger, M. (2011). Post-error adjustments. *Frontiers in Psychology*, *2*, 233. <http://doi.org/10.3389/fpsyg.2011.00233>
- Donner, T. H., Siegel, M., Fries, P., & Engel, A. K. (2009). Buildup of Choice-Predictive Activity in Human Motor Cortex during Perceptual Decision Making. *Current Biology*, *19*(18), 1581–1585. <http://doi.org/10.1016/j.cub.2009.07.066>
- Hawkins, G. E., Forstmann, B. U., Wagenmakers, E. J., Ratcliff, R., & Brown, S. D. (2015). Revisiting the Evidence for Collapsing Boundaries and Urgency Signals in Perceptual Decision-Making. *Journal of Neuroscience*, *35*(6), 2476–2484. <http://doi.org/10.1523/JNEUROSCI.2410-14.2015>
- Pape, A.-A., & Siegel, M. (2016). Motor cortex activity predicts response alternation during sensorimotor decisions. *Nature Communications*, *7*, 1–10. <http://doi.org/10.1038/ncomms13098>
- Stetson, C., & Andersen, R. A. (2014). The Parietal Reach Region Selectively Anti-Synchronizes with Dorsal Premotor Cortex during Planning. *Journal of Neuroscience*, *34*(36), 11948–11958. <http://doi.org/10.1523/JNEUROSCI.0097-14.2014>
- Zhang, Y., Chen, Y., Bressler, S. L., & Ding, M. (2008). Response preparation and inhibition: The role of the cortical sensorimotor beta rhythm. *Neuroscience*, *156*(1), 238–246. <http://doi.org/10.1016/j.neuroscience.2008.06.061>

Reviewers' comments:

Reviewer #1 (Remarks to the Author):

In my view, the authors have done an excellent job of addressing the comments raised. The additional of the computational model has certainly strengthened the interpretation of the data and the correspondences between the model simulations and neural data are striking. I have just two minor comments

1. Why did the authors choose to implement this particular DDM model? I am aware of at least two pre-existing DDM variants that are designed to take account of flanker-type effects - the Dual-Stage Two-Phase Model of Selective Attention (Hubner et al 2010) and the Shrinking Spotlight Model (White et al 2011). It would be important to give the reader a sense of why these models were not considered or how similar/different the chosen model is to these other variants.

2. I am still somewhat confused as to why the authors are concluding that BPL and not contralateral beta reflects the 'decision threshold'. I get the impression it may boil down to their observations on lines 271 and 273 that "we found that higher RT ($t_{862} = 13.4$, $p < 10^{-35}$) per trial was associated with decreased BPL at response (Supplementary Fig. 7e). Neither ipsilateral ($t_{862} = -3.6$, $p < 10^{-3}$) nor contralateral hemispheres ($t_{862} = 5.1$, $p < 10^{-5}$) reflected this effect alone." First, the phrasing is ambiguous, I do not know what the authors mean by 'neither...reflected this effect alone'. The statistical tests indicate significant effects of RT on both ipsi and contra beta activity. Furthermore, it would be important to show waveforms separating contra and ipsi beta as a function of RT so that these effects can be visualised. Although this issue is rather peripheral to the main thrust of the paper I think it's an important point to clarify for researchers seeking to use these kinds of signals in the future. It certainly seems clear that BPL is sensitive to MODULATIONS of the decision threshold, but whether lateralisation of motor activity itself must hit a threshold for decisions to be reported is a separate and important question. For example, if ipsi and contra beta reflect the output of racing accumulators, then one would not always expect the lateralisation index to reflect the threshold but rather the amplitude of the winning accumulator (i.e. beta contralateral to the ultimately chosen response).

Reviewer #2 (Remarks to the Author):

The author's revisions complete the necessary background for the impressive and important conclusions they draw from these findings.

Reviewer #3 (Remarks to the Author):

I applaud the authors for their thorough revision, additional analyses and results, which substantially strengthen the paper. In particular, including the DDM renders the paper much stronger. The authors have successfully addressed all my concerns.

We are very pleased with all reviewer's positive assessments of our revision. Below we respond to the two points raised by reviewer one. Again, we marked changes in the manuscript in blue print and reproductions of the manuscript in red.

Reviewer #1

In my view, the authors have done an excellent job of addressing the comments raised. The additional of the computational model has certainly strengthened the interpretation of the data and the correspondences between the model simulations and neural data are striking. I have just two minor comments

Thank you for this assessment. We respond to the two minor comments below and have updated the manuscript accordingly.

1. Why did the authors choose to implement this particular DDM model? I am aware of at least two pre-existing DDM variants that are designed to take account of flanker-type effects - the Dual-Stage Two-Phase Model of Selective Attention (Hubner et al 2010) and the Shrinking Spotlight Model (White et al 2011). It would be important to give the reader a sense of why these models were not considered or how similar/different the chosen model is to these other variants.

We have added a comparison to the methods that outlines our reasons for designing our model and which includes a description of similarities and differences to the two previously used DDM variants in the context of the flanker task. In short, both models are very similar to our implementation. Yet, as we had to implement a slightly more complex model simulating all variance parameters in order to allow a valid comparison against the neural signal (which naturally contains variance), we implemented the (in our view) most simple version of a DDM for this task that only assumes that attention shifts from flankers to the target once the target is on screen and that these two stages of information processing can differ from each other (f) which should be different on each trial (sf). Therefore, we did not have to simulate two processes (stimulus and response selection) nor to assume a shrinking function for the attentional spotlight. Note that our task design differs slightly from the designs used by Hubner et al. (2010) and White et al. (2011) such that flanker stimuli alone precede the presentation of the full target-flanker stimulus array which renders our simplification plausible. Both previous models would lead to very similar results behaviourally (compared to our model), but would not allow the same simplicity in comparing post-error and post-correct conditions as well as the neural time-course.

We reproduce the new section below which is now included at p. 14 of the revised manuscript.

Two similar variants of DDMs have been implemented to model behavioural effects in conflict tasks that both assume modulations of visual selectivity in multiple stages^{58,60}. In accordance with these previous reports, we found that multiple stages (reflecting differences in selective attention) are required to describe the behavioural data adequately. However, in order to keep the model as simple as possible, we assumed that the stages are defined by the time stimuli are presented on screen rather than two separate processes reflecting stimulus and response selection⁶⁰ or a spotlight that changes over time and reduces the influence of flanker stimuli⁵⁸. Thus, our model is very similar to these previous DDM versions and provides an easy means to comparing modelled and neural signal time-courses as well as parameter values representing attentional modulations between conditions.

2. I am still somewhat confused as to why the authors are concluding that BPL and not contralateral beta reflects the 'decision threshold'. I get the impression it may boil down to their observations on lines 271 and 273 that "we found that higher RT ($t_{862} = 13.4, p < 10^{-35} 271$) per trial was associated with decreased BPL at response (Supplementary Fig. 7e). Neither ipsilateral ($t_{862} = -3.6, p < 10^{-3}$) nor contralateral hemispheres ($t_{862} = 5.1, p < 10^{-5}$) reflected this effect alone." First, the phrasing is ambiguous, I do not know what the authors mean by 'neither...reflected this effect alone'. The statistical tests indicate significant effects of RT on both ipsi and contra beta activity. Furthermore, it would be important to show waveforms separating contra and ipsi beta as a function of RT so that these effects can be visualised. Although this issue is rather peripheral to the main thrust of the paper I think it's an important point to clarify for researchers seeking to use these kinds of signals in the future. It certainly seems clear that BPL is sensitive to MODULATIONS of the decision threshold, but whether lateralisation of motor activity itself must hit a threshold for decisions to be reported is a separate and important question. For example, if ipsi and contra beta reflect the output of racing accumulators, then one would not always expect the lateralisation index to reflect the threshold but rather the amplitude of the winning accumulator (i.e. beta contralateral to the ultimately chosen response).

We agree with the reviewer and have made it clearer that especially modulations of the threshold are compatible with changes in BPL. Additionally, we have rephrased the section cited above and made it clear that this modulation is more strongly (yet not exclusively) reflected in BPL compared to contralateral beta power alone. Finally, we have added a plot of ipsilateral and contralateral beta power time-courses split up by slow and fast responses to Supplementary Fig. 7g,h that displays the modulation over the contralateral and ipsilateral hemispheres separately.

We wish to clarify that we do not exclude that separate racing accumulators are a valid model of speeded decision-making under conflict, but we demonstrate that the reduction that is assumed by the DDM and which allows it to easily model

mutually exclusive decisions by assuming that evidence for response A counteracts evidence for response B, is valid and compatible with the neural signal reflected in BPL. It will be highly interesting to design tasks that can clearly dissociate between race models and the DDM in future studies via contrasting BPL and contralateral beta power. We acknowledge this in the discussion.

All relevant sections are reproduced below:

Main manuscript p. 7:

Additionally, we found that higher RT ($t_{862} = 13.4, p < 10^{-35}$) per trial was associated with decreased BPL at response (Supplementary Fig. 7e). This effect was stronger than the effect on either hemisphere alone (Supplementary Fig. 7g-h, contrast contralateral against BPL effect $t_{862} = 5.3, p < 10^{-6}$, main effects contralateral $t_{862} = 5.1, p < 10^{-5}$ and ipsilateral $t_{862} = -3.6, p < 10^{-3}$). Reduced BPL would be compatible with the notion of dynamic decision boundaries⁴⁸, which adaptively decrease under time-pressure or urgency. Thus, threshold modulations appear to be reflected especially in BPL.

Main manuscript discussion in p. 9:

These results do not rule out that responses are finally determined by two separate accumulation processes in both hemispheres. However, these findings suggest that at least in the context of mutually exclusive decisions DDM assumptions are neurally validated and BPL appears to be an especially sensitive measure of response threshold modulations.

Figure S7e-h.

As in the stimulus-locked analyses, we find that in BPL (e) as well as modelled signal (f) a lateralisation towards the given response is pronounced for fast responses. Notably, BPL at response execution is reduced for later responses. If BPL is seen as a reflection of response thresholds, this observation would be compatible with dynamically changing decision bounds within a trial¹⁰.

This effect is partly explained by a reduction of beta power over the contralateral hemisphere (g), while on average beta power over the ipsilateral hemisphere is barely changed when responses are given later in a trial (h). Yet, BPL showed a stronger influence of response time than either of the hemispheres alone suggesting that it may be an especially valid marker of response threshold modulations.

REVIEWERS' COMMENTS:

Reviewer #1 (Remarks to the Author):

The authors have addressed all of my comments. I am glad that this nice paper will be published at Nat Comms